# AGENTVQA: A UNIFIED BENCHMARK FOR AGENTIC VISUAL UNDERSTANDING

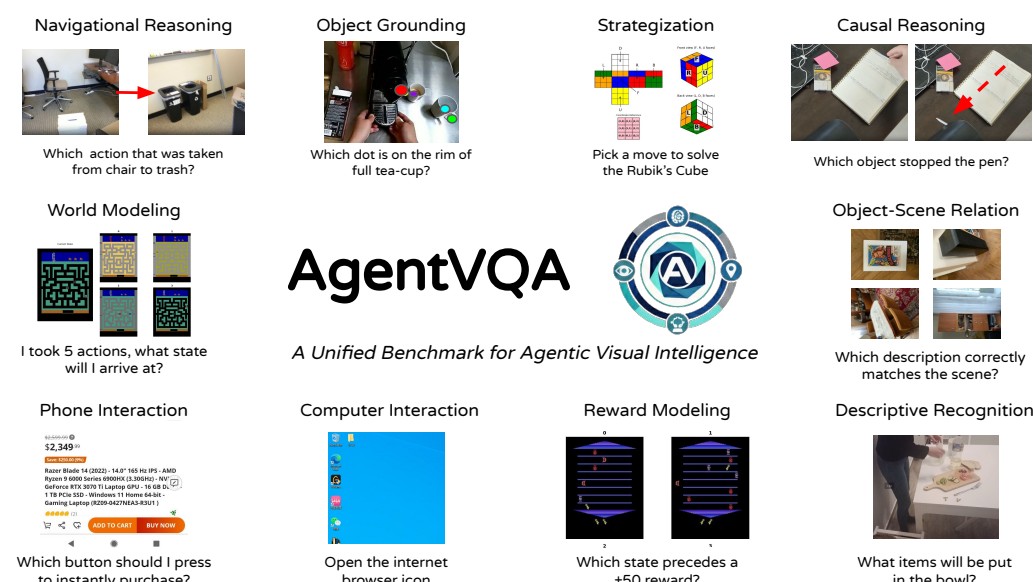

Figure 1: **The AgentVQA Benchmark.** AgentVQA unifies 14 challenging datasets across five domains: Web Agents, Egocentric Videos, Robotics, Games, and Spatial Understanding, into a standard MCQ format. These tasks are diverse with some using action-histories to test episodic memory and others using video inputs to evaluate temporal understanding.

## ABSTRACT

Vision-language models (VLMs) can perform a broad range of tasks across diverse settings. Yet their performance in agentic contexts remains poorly understood. Existing benchmarks are domain-specific, making comprehensive evaluation difficult, and they often require compute-expensive online simulators. To address this gap, we introduce AgentVQA, a benchmark for systematically evaluating agentic capabilities in VLMs. AgentVQA offers three key advantages: (1) *Comprehensive* – it consists of 14 datasets spanning five critical agentic domains: Web Agents, Robotics, Egocentric Videos, Games, and Spatial Understanding. (2) *Standardized* – we reformulate diverse tasks, like trajectory-based web navigation and gameplay, into a unified multiple-choice question (MCQ) format. We balance the sample distribution across multiple domains, data formats, and semantic categories. (3) *Challenging* – our data processing pipeline generates hard negative options in MCQs, which are then manually reviewed for correctness. Among all the models we evaluate, the best achieves a mere ∼60% accuracy. Furthermore, our ablation studies highlight key error modes where current VLMs can be improved.

## 1 INTRODUCTION

Vision-Language Models (VLMs) are quickly becoming the decision-making core for agentic systems spanning robotics, wearable assistants, web navigation, or gameplay. With only prompting or minimal

fine-tuning, VLMs often outperform their domain-specific counterparts (Koh et al., 2024; Black et al., 2024; Zhou et al., 2025; Zhang et al., 2025; Sarch et al., 2025). This approach promises a scalable path to universal agentic behavior. However, our understanding of the real-world agentic capabilities of these models is still limited.

In other domains like math or coding, standardized benchmarks systematically track model capabilities and limitations (Jimenez et al., 2023; Balunović et al., 2025). For agents, online evaluations in real or simulated environments remain the gold standard for assessing agentic performance (Yao et al., 2024), but due to computational cost and reproducibility, they present significant practical challenges for rapid, large-scale model comparison (Henderson et al., 2018; Dasari et al., 2022).

As a result, VLMs' agentic capabilities lack comparable evaluation frameworks and remain poorly understood. State-of-the-art models are often assessed on general-purpose benchmarks like MMMU (Yue et al., 2024), MMBench (Liu et al., 2024) and GPQA (Rein et al., 2024). While these track broad visual-language understanding, the questions and visual inputs do not reflect agentic decision-making scenarios (visualized in Figure 2). Recent works (Yao et al., 2024; Yehudai et al., 2025; Wong et al., 2025) show that performance on general-purpose VQA benchmarks does not reliably correlate with success in agentic tasks.

To address this limitation, several domain-specific agentic benchmarks have been proposed. These benchmarks for agentic tasks have made valuable contributions to their respective domains (Majumdar et al., 2024; Cheng et al., 2024a; Chen et al., 2025). However, they remain insufficient for evaluating generalist agents, because they are fragmented across domains. They fail to provide comprehensive, unified evaluations of the diverse domains and inputs expected from generalist agents.

Prior works, such as Li et al. (2023a), have made significant efforts in aggregating diverse, multi-image tasks into a standardized format. Yet, their curation for agent-oriented tasks focuses on general image understanding and classification of actions in videos. They largely avoid interactive decision-making data required by agents in real-world, complex environments. This highlights the need for a robust and standardized offline evaluation framework to track progress in agentic AI.

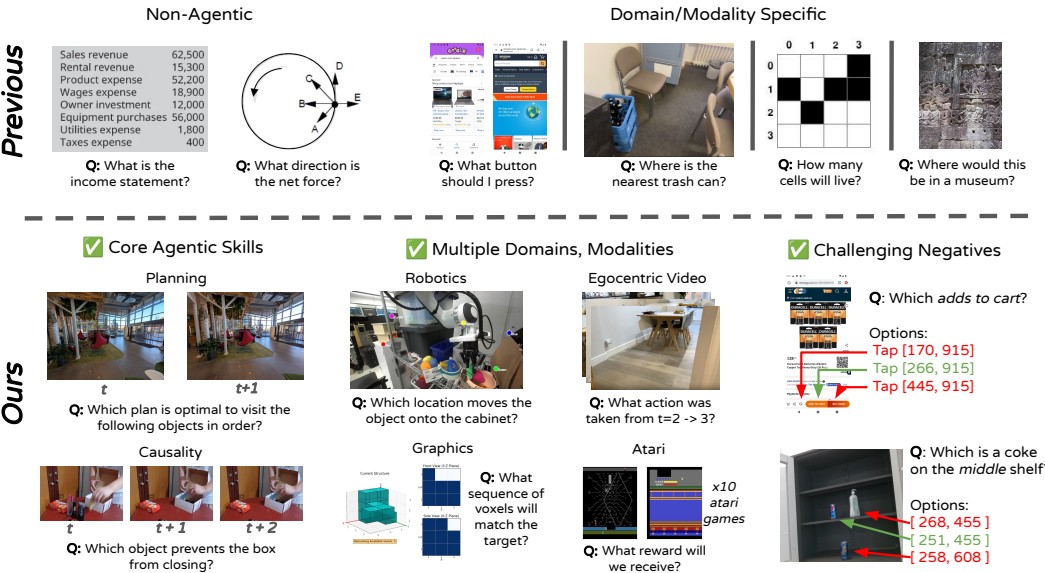

Figure 2: **Comparison between AgentVQA and previous benchmarks.** Examples above the dashed line illustrate questions from existing VLM benchmarks, which are either non-agentic (Yue et al., 2024) or domain-specific (Deng et al., 2023). AgentVQA addresses three evaluation gaps: (1) it tests core agent skills by unifying the evaluation of agent tasks, which were previously isolated in separate datasets; (2) it spans multiple domains and modalities, evaluating performance across diverse domains, image/video modalities, and action trajectories; and (3) it incorporates challenging negatives through distractors, including semantically similar and near-miss actions.

We introduce **AgentVQA**, a **comprehensive** benchmark designed to address these limitations and enable a systematic analysis of VLM capabilities for agentic tasks. AgentVQA unifies 14 challenging datasets across five domains: Web Agents, Egocentric Videos, Robotics, Games, and Spatial Understanding. These domains are chosen for their relevance to the current frontiers of agentic AI research (visualized in Figure 1). AgentVQA consists of 13,400 MCQs spanning 18,400 images and 2,000 videos from pre-existing trajectory and VQA datasets. Our analysis reveals substantial rank shifts between AgentVQA and MMMU, GPQA-Diamond, and OpenEQA (Table 4), indicating that the model rankings in our benchmark are largely uncorrelated with those in both general-purpose VQA and domain-specific benchmarks.

AgentVQA ensures **standardization** by filtering and transforming dynamic tasks, such as gameplay and trajectory-based web navigation, into a unified MCQ format. For instance, we transform gameplay datasets with multiple valid next steps into nuanced reward modeling questions that challenge a model's understanding of optimal strategies. Similarly, we transform web trajectories by pairing the ground-truth action with other plausible but incorrect grounding options.

AgentVQA is a **challenging** benchmark. This is ensured by our methodology for generating plausible "hard negatives" distractor options. These distractors fall into two main categories: near-miss actions (e.g., clicking nearby but not on the correct button) and semantically similar options (e.g., choosing apple instead of orange). The generation strategy itself is validated through an extensive initial analysis, followed by manual verification to ensure each distractor is both plausible and unambiguously incorrect. The benchmark's resulting difficulty is reflected in our findings: even the top-performing model, GPT-5 (thinking-high), achieves an overall accuracy of only ∼60%.

AgentVQA provides data-driven insights into the current state of the agentic capabilities of VLMs. Our evaluation across 15 open and closed-source models of varying sizes leads to a detailed analysis. For instance, top-performing models in one category do not always perform best in other categories. While GPT-5 (thinking-high) dominates in Spatial Understanding (72%), Games (60%), and Egocentric Videos (70%), Qwen2.5-VL (72B) leads in Web Agents (57%) and Robotics (52%).

Furthermore, our error mode analysis, based on a manually crafted and verified categorization of reasoning outputs, reveals highly domain-specific failure patterns. The most prevalent error in Web Agents is grounding errors (46%). Spatial reasoning failures dominate other domains, with spatial confusion the top error in both Robotics (51%) and Egocentric Videos (35%), and spatial misconstruction in Spatial Understanding (28%). Finally, the errors in Games are largely high-level reasoning breakdowns (40%), pinpointing a clear split between low-level perceptual grounding and high-level abstract reasoning as the fundamental challenges in agentic tasks.

## 2 RELATED WORK

**VLM as agents.** Recent advances in multimodal AI have been driven by the development of Vision-Language Models (VLMs). These models, such as QwenVL (Bai et al., 2023), GPT-4o (Hurst et al., 2024), and Gemini (Comanici et al., 2025), are typically pretrained on vast, web-scale datasets of paired images and text. This equips them with a broad general-purpose understanding of both visual concepts and natural language, making them an ideal compute unit for agents. Several works have successfully adapted VLMs into specific domains, either by fine-tuning or scaffolding pre-trained models (Black et al., 2024; Koh et al., 2024; Zhou et al., 2025; Zhang et al., 2025; Sarch et al., 2025). However, this begs the question: how do we evaluate agentic capabilities *across domains*?

**General visual question answering.** A wealth of benchmarks have been developed to assess the general capabilities of VLMs, including SEED-Bench (Li et al., 2023a), MMBench (Liu et al., 2024), or MMMU (Yue et al., 2024). While invaluable for measuring core competencies and reasoning, they are ill-suited as proxies for agentic intelligence. Agentic tasks often require episodic memory (Majumdar et al., 2024) because either the modalities are iterative, e.g., a stream of image observations, or the task requires interactive reasoning, e.g., tool-calling. Yao et al. (2024) found that GPT-4o, despite excelling on traditional benchmarks, achieves a failure rate less than 50% on realistic customer service tasks requiring multi-turn interactions. In Fig. 4, we confirm this observation by the divergence between rankings in AgentVQA vs MMMU and GPQA-Diamond.

**Agentic benchmarks.** The agentic evaluation landscape is bifurcated into language-only benchmarks, e.g., SWE-bench (Jimenez et al., 2023) or AgentBench (Liu et al., 2023), that test code generation

and tool use, and vision-language benchmarks that require visual grounding and spatial reasoning. *AgentVQA is the latter.* Additionally, benchmarks are further split by execution pattern: online versus offline. Online benchmarks, e.g., (Savva et al., 2019; Fan et al., 2022; Li et al., 2023b; Koh et al., 2024; Zhang et al., 2025), are ideal because performant agents must handle non-deterministic environment interactions, which is fundamental to real-world deployment.

However, online evaluation can be time-consuming or require significant infrastructure (Henderson et al., 2018; Du et al., 2024), so offline agentic benchmarks have grown in popularity. Offline benchmarks, e.g., (Rawles et al., 2023; Yang et al., 2025a; Team et al., 2025; Tong et al., 2025; Gong et al., 2025), consist of verifiable question-answer tasks (often MCQ) paired with visual inputs. Existing offline agentic benchmarks are domain-specific, and the fragmented state of offline agentic evaluation makes it difficult to assess cross-domain visual generalization. (we outline a full comparison between canonical offline and online VLM datasets in Table 5). Aggregating performance across datasets is also problematic because they consist of non-uniform answer formats, varying degrees of answer choice "hardness", and varying sample distributions. This makes it difficult to distinguish skills needed for cross-dataset or cross-domain generalization.

AgentVQA addresses these limitations by 1) curating relevant datasets across domains. 2) filtering and transforming representative subsets of the datasets into a standardized MCQ format focused on hard negatives. 3) annotating category metadata to understand cross-dataset skill distributions. In the following sections, we expand on each of these steps, as well as our evaluation on VLMs.

## 3 THE AGENTVQA BENCHMARK

To systematically analyze the agentic capabilities of general-purpose VLMs, a new evaluation framework is required. We define an agentic task as one requiring a model to perceive its environment, reason about dynamics, and select actions to achieve a goal (Bengio et al., 2025). Unlike passive VQA, agentic perception is instrumental, a prerequisite for downstream action (Reed et al., 2022). We provide a detailed taxonomy of these agentic skills in Appendix C. Existing benchmarks fall short, either by focusing on single domains (Majumdar et al., 2024; Cheng et al., 2024a), which prevents a holistic assessment, or by testing general VQA skills that are poor proxies for agentic competence (Yue et al., 2024; Rein et al., 2024). AgentVQA answers a central research question: where do modern, general-purpose VLMs succeed or fail across a broad range of agentic tasks? Its construction follows three core principles: **comprehensive** domain coverage to test for skill generality, a **standardized** MCQ format for scalable analysis, and the use of systematically generated hard negatives to ensure it is **challenging**.

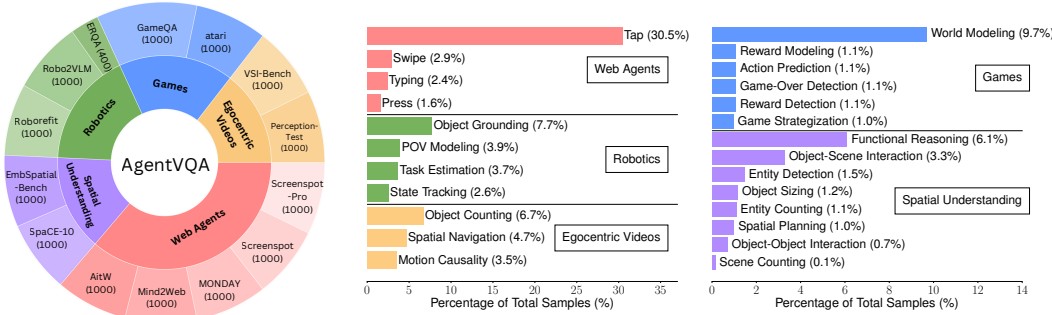

Figure 3: **Distribution of the 13,400 questions in the AgentVQA benchmark. Left:** The distribution across the 14 source datasets (with sample counts in parentheses), organized within the five core agentic domains. **Right:** A complementary view illustrating the sample distribution across our 25 defined sub-task categories.

### 3.1 DATASET OVERVIEW

An overview of our benchmark domains and evaluation types is summarized in Figure 3. AgentVQA is a large-scale, multi-domain benchmark. It comprises 13,400 standardized MCQ questions designed to probe the agentic intelligence of VLMs. The questions are curated from 14 diverse datasets spanning: Web Agents, Egocentric Videos, Robotics, Games, and Spatial Understanding.

Our domains were chosen to express distinct agentic interactions. These include digital interaction and GUI navigation (Web Agents) to embodied perception in the physical world (Robotics, Egocentric Videos), strategic decision-making (Games), and foundational spatial awareness (Spatial Understanding). We use agentic-relevance, public availability, MCQ-format potential, and verifiability as our selection criteria (more details in Appendix A).

A core feature of AgentVQA is its task diversity. To ensure multifaceted evaluation, questions are structured around 25 distinct sub-task categories (Figure 3). Our categories encompass a wide range of agentic skills, ranging from low-level action prediction (e.g., Tap/Click, Type) and spatial grounding to more complex trajectory-based reasoning, spatiotemporal understanding, and state-value estimation (e.g., Reward Modeling). Our categorization starts with manual inspection of raw samples to identify the most informative question types across datasets, then programmatically mapping each sample to a category (more details in Appendix D).

Unlike benchmarks focused solely on static images, 15% of our questions require reasoning over dynamic video clips from the Egocentric Videos. AgentVQA also includes a high concentration of sequential tasks - 26% of all questions are trajectory-based. So for either modality, these questions provide a crucial test of spatiotemporal understanding. AgentVQA's diversity in domain, skill-categorization, and modality promote granular analysis of model capabilities. The overall distribution of questions across these axes is shown in Figure 3.

## 3.2 DATA COLLECTION

We chose datasets ripe with the following characteristics: agentic signal, domain/modality coverage, offline convertibility, availability, and scale. The full process is described in Appendix A, and this process yielded the following source datasets: AitW (Rawles et al., 2023), MONDAY (Jang et al., 2025), Mind2Web (Deng et al., 2023), Screenspot (Cheng et al., 2024a) and Screenspot-pro (Li et al., 2025) for Web Agents; ERQA (Team et al., 2025), Robo2VLM (Chen et al., 2025), and Roborefit (Lu et al., 2023) for Robotics; VSI-Bench (Yang et al., 2025a) and Perception-Test (Patraucean et al., 2023) for Egocentric Videos; GameQA (Zhang et al., 2025) and Atari (Zhang et al., 2020) for Games; SpaCE-10 (Gong et al., 2025) and EmbSpatial-Bench (Du et al., 2024) for Spatial Understanding.

We additionally show the composition of AgentVQA in Figure 3. To ensure a manageable yet representative benchmark, we subsample 1000 instances from each source dataset (except ERQA with 400). This is based on the smallest sample size that consistently yields <1% standard deviation in model performance (more details in Appendix E). Our curation yields 13400 questions, spanning 18400 images and 2000 videos.

## 3.3 DATA ANNOTATION AND MCQ CONVERSION

To standardize the outputs and employ systematic evaluation, we convert all questions into multiple choice questions (MCQs). For the datasets which aren't already in MCQ format, we systematically create two types of hard negatives. These are near-miss negatives (e.g., in Web Agents, clicking a few pixels away from the correct UI element) and semantically similar negatives (e.g., In Robotics, choosing coordinates of a mobile phone instead of a tablet.). We summarize our multiple choice conversion process below, with additional details in Appendix B.

**Generating negatives.** We generate our negatives using a VLM-assisted pipeline with Gemini 2.5 Pro. The model receives context to assist generation including the visual input with ground-truth annotations overlaid, the task prompt, correct answer, action history for sequential tasks, and relevant metadata (e.g., coordinate formats, action types). We prompt the VLM to generate three hard negatives per question with a mix of near-miss and semantically similar options.

**Quality control.** To ensure accurate and challenging negatives, the generated negatives undergo quality control. First, for grounding tasks with available bounding boxes, we automatically filter out any negatives whose coordinates fall within the ground-truth region. Second, human annotators manually review a subset of samples for each domain to ensure the generated negatives are both plausible (could realistically confuse a model) and unambiguously incorrect (maintaining a single correct answer). This two-stage verification ensures our MCQs are accurate and challenging for fine-grained understanding.

| Model | Web Agents | | | | | | Robotics | | | |
|---|---|---|---|---|---|---|---|---|---|---|
| | AitW ++ | MONDAY ++ | Mind2Web ++ | Screenspot ++ | Screenspot-pro ++ | Avg | ERQA | Robo2VLM + | Roborefit ++ | Avg |
| GPT-5 thinking-high 🧑‍🔬🔒 | **61.9** | **51.6** | **72.2** | 48.5 | 41.9 | 55.2 | **58.9** | 46.3 | 48.3 | 51.2 |
| GPT-5 thinking-min 🔒 | 53.5 | 49.8 | 68.1 | 40.3 | 39.8 | 50.3 | 57.0 | 44.8 | 29.2 | 43.7 |
| Gemini 2.5 Pro 🔒 | 50.3 | 36.7 | 63.0 | 34.8 | 43.0 | 45.6 | 47.0 | 40.7 | 32.8 | 40.2 |
| GPT-4o 🔒 | 40.5 | 29.6 | 57.9 | 35.1 | 39.4 | 40.5 | 40.7 | 32.1 | 33.5 | 35.4 |
| GLM-4.1V-Thinking 🧑‍🔬 | 42.0 | 33.5 | 63.4 | 30.6 | 42.2 | 42.3 | 43.3 | 35.7 | 24.8 | 34.6 |
| GLM-4.1V-Base | 36.5 | 33.4 | 63.6 | 28.8 | 39.3 | 40.3 | 30.3 | 31.9 | 35.0 | 32.4 |
| Kimi-VL-Thinking 🧑‍🔬 | 41.5 | 31.4 | 29.79 | 29.5 | 33.5 | 33.1 | 34.0 | 40.9 | 35.7 | 36.9 |
| Kimi-VL-Instruct | 37.2 | 27.6 | 24.0 | 32.2 | **50.4** | 34.3 | 34.3 | 38.8 | 39.1 | 37.4 |
| Phi-4 Multimodal | 33.2 | 35.0 | 50.0 | 21.7 | 28.5 | 33.7 | 38.8 | 35.9 | 23.2 | 32.6 |
| Llama 4 Scout | 42.7 | 36.9 | 57.7 | 32.3 | 41.0 | 42.1 | 37.0 | 29.4 | 37.8 | 34.7 |
| Llama 4 Maverick | 38.8 | 35.6 | 54.1 | 33.1 | 36.1 | 39.6 | 35.8 | 32.4 | 39.1 | 35.8 |
| Qwen2.5-VL (3B) | 37.3 | 38.5 | 51.5 | 55.1 | 40.6 | 44.6 | 32.5 | 31.3 | 49.8 | 37.9 |
| Qwen2.5-VL (7B) | 54.2 | 44.7 | 56.9 | 59.6 | 42.1 | 51.5 | 36.0 | 42.8 | 54.4 | 44.4 |
| Qwen2.5-VL (32B) | 47.5 | 50.6 | 62.8 | 73.8 | 40.2 | 55.0 | 39.0 | 43.0 | 61.1 | 47.7 |
| Qwen2.5-VL (72B) | 51.4 | 48.8 | 65.5 | **76.5** | 44.5 | **57.3** | 39.5 | **50.0** | **65.0** | **51.5** |

| Model | Egocentric Videos | | | Games | | | Spatial Understanding | | |
|---|---|---|---|---|---|---|---|---|---|
| | VSI-Bench + | Perception-Test + | Avg | GameQA + | atari ++ | Avg | SpaCE-10 + | EmbSpatial-Bench + | Avg |
| GPT-5 thinking-high 🧑‍🔬🔒 | **65.7** | 75.0 | **70.4** | 64.9 | **55.4** | **60.2** | **59.8** | **83.2** | **71.5** |
| GPT-5 thinking-min 🔒 | 57.9 | **78.1** | 68.0 | **66.0** | 28.7 | 47.4 | 54.8 | 78.1 | 66.5 |
| Gemini 2.5 Pro 🔒 | 43.3 | 65.4 | 54.4 | 32.1 | 46.8 | 39.5 | 50.1 | 68.9 | 59.5 |
| GPT-4o 🔒 | 33.3 | 56.4 | 44.9 | 26.5 | 36.3 | 31.4 | 42.5 | 59.7 | 51.1 |
| GLM-4.1V-Thinking 🧑‍🔬 | 32.6 | 56.6 | 44.6 | 41.0 | 29.4 | 35.2 | 48.0 | 76.3 | 62.2 |
| GLM-4.1V-Base | 25.3 | 35.7 | 30.5 | 21.1 | 25.2 | 23.2 | 33.9 | 76.1 | 55.0 |
| Kimi-VL-Thinking 🧑‍🔬 | 26.9 | 31.0 | 29.0 | 20.3 | 27.7 | 24.0 | 30.6 | 71.0 | 50.8 |
| Kimi-VL-Instruct | 39.8 | 52.0 | 45.9 | 25.2 | 31.0 | 28.1 | 44.3 | 57.9 | 51.1 |
| Phi-4 Multimodal | 41.2 | 70.6 | 55.9 | 29.2 | 24.0 | 26.6 | 45.5 | 70.5 | 58.0 |
| Llama 4 Scout | 38.6 | 51.8 | 45.2 | 28.9 | 35.9 | 32.4 | 44.1 | 54.3 | 49.2 |
| Llama 4 Maverick | 36.0 | 55.4 | 45.7 | 25.8 | 31.2 | 28.5 | 42.0 | 62.6 | 52.3 |
| Qwen2.5-VL (3B) | 37.2 | 62.1 | 49.7 | 24.3 | 25.2 | 24.8 | 30.0 | 60.3 | 45.2 |
| Qwen2.5-VL (7B) | 37.4 | 65.8 | 51.6 | 31.5 | 26.7 | 29.1 | 38.9 | 71.6 | 55.3 |
| Qwen2.5-VL (32B) | 40.5 | 62.2 | 51.4 | 35.8 | 29.3 | 32.6 | 45.5 | 74.6 | 60.1 |
| Qwen2.5-VL (72B) | 38.5 | 64.5 | 51.5 | 40.5 | 40.4 | 40.5 | 49.0 | 73.0 | 61.0 |

Table 1: **Performance of VLMs on AgentVQA across the five agentic domains by dataset.**
🧑‍🔬 represents Reasoning Models. 🔒 represents closed-source models. Bolded and underlined numbers indicate best and second-best rank per column, respectively. + denotes a filtered dataset, while ++ denotes a filtered and transformed dataset..

## 4 EXPERIMENTS

This section details our experimental setup, main performance results, in-depth analyses of model behaviors, and strengths/weaknesses across a variety of agentic dimensions.

### 4.1 EXPERIMENTAL SETUP

**Models.** We evaluate a diverse suite of 15 prominent VLMs to ensure a comprehensive assessment of the current landscape. Our selection includes closed-source, proprietary models: GPT-5 thinking-high/minimal (OpenAI, 2025), GPT-4o (Hurst et al., 2024), and Gemini 2.5 Pro (Comanici et al., 2025). We also include a wide range of open-source models: GLM-4.1V-Thinking/Base (Hong et al., 2025), Kimi-VL-A3B-Thinking/Instruct (Du et al., 2025) (for brevity we shorten Kimi-VL-A3B-Thinking-2506 to *Kimi-VL-Thinking*), Llama 4 Scout (Meta, 2025), Llama 4 Maverick (Meta, 2025), Phi-4-multimodal (Abdin et al., 2024), and the Qwen2.5-VL series (3B, 7B, 32B, and 72B) (Bai et al., 2025).

Models are categorized as "Reasoning" or "Non-Reasoning" based on their generation of intermediate thinking tokens. A complete list of evaluated models is provided in Table 1.

**Evaluation setup.** For models that do not natively support video input, we sample 32 frames at evenly spaced intervals. This number is validated by our own ablation studies and prior work (Yang et al., 2025a). Models are evaluated with a temperature of 0.8 and 0.2 for reasoning and non-reasoning models, respectively. For models without manual temperature (e.g., GPT-5), we use the default.

We use a standard, simple prompt structure that provides the question and any relevant context. This prompt queries the model to output only the single letter corresponding to its choice. Final answers are extracted programmatically using a set of parsing rules. All visual inputs are provided at their native resolution without resizing. Further details on the exact prompts used and the answer extraction logic are available in Appendices L and F.

## 4.2 OVERALL PERFORMANCE

**Top VLMs struggle with AgentVQA.** The averaged accuracy of each model is presented in Table 1. The results reveal a significant absolute performance gap across the board. The average performance across all 15 models ranges from 34% in Games to 53% in Spatial Understanding.

Even the top performing model, GPT-5 thinking-high, achieves an overall accuracy of only 60% across all domains. This underscores the difficulty of our benchmark. A clear tiering is observable: the largest proprietary models (GPT-5, Gemini 2.5 Pro) are at the top. This is followed by the largest dense, open-source models, Qwen2.5-VL 32B and 72B, then followed by the smaller, open-source models comprise the lower tiers. Among the five domains, models perform best on Spatial Understanding (avg. 53%) followed by Egocentric Videos (avg. 49%), Web Agents (avg. 44%), Robotics (avg. 40%) and worst on Games (avg. 34%). This indicates that tasks requiring passive observation are currently more tractable than those demanding active, strategic planning.

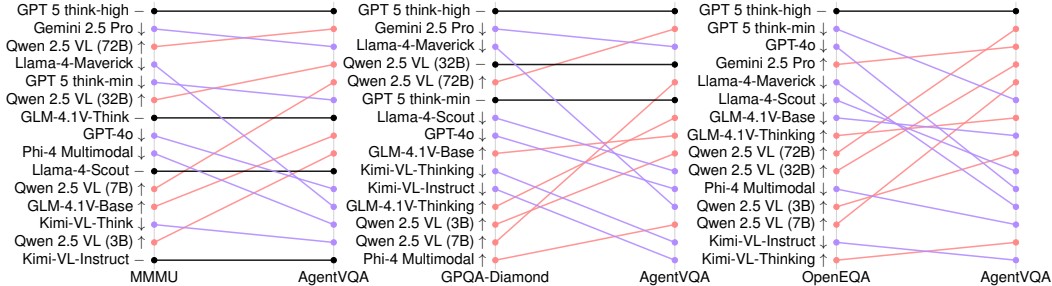

Figure 4: **Divergence in model performance rankings between AgentVQA and three existing benchmarks.** Lines are colored red if a model's rank improves and purple if it degrades. The rank divergence shows that our benchmark is largely uncorrelated with general VQA and domain-specific agentic benchmarks. While a few models maintain their ranks, the relative positions of most other models change significantly. Notably, in the comparison with OpenEQA, only one model maintains its relative ranking.

## 4.3 RANK CORRELATION

**Low correlation with existing benchmarks.** To validate the necessity of a comprehensive and specialized agentic benchmark, we compare the performance rankings of models on AgentVQA to their rankings on both general-purpose VQA and Reasoning benchmarks (MMMU (Yue et al., 2024) and GPQA-Diamond (Rein et al., 2024)) and a domain-specific agentic benchmark (Open-EQA (Majumdar et al., 2024)). We ran our own evaluations with standardized prompts to ensure consistent and comparable model accuracies.

As shown in Figure 4, the model rankings of AgentVQA differ drastically to all. This finding is also quantitatively confirmed by low Spearman's rank correlation coefficients against MMMU ($\rho \approx 0.69, p \approx 0.004$), GPQA-Diamond ($\rho = 0.52, p = 0.046$), and even Open-EQA ($\rho \approx 0.44$). Additionally, Open-EQA uses GPT-4 as an evaluator so ground-truth results are unreliable compared to the deterministic comparison put forth by AgentVQA, MMMU, or GPTQA-Diamond.

The relative ranking of the top model (GPT-5 thinking-high) remains consistent. However, other open-source models like Qwen2.5-VL series, across various sizes, are more performant on AgentVQA, while LLama 4 models worsen. In the next section, we introduce ablation studies that help explain factors underlying performance differences.

**Correlation with Online Agents.** To address concerns that offline MCQs may not predict online performance, we compared AgentVQA rankings against success rates on interactive benchmarks OSWorld (Xie et al., 2024) and VideoGameBench (Zhang et al., 2025). On VideoGameBench, rankings for Gemini-2.5-Pro, GPT-4o, and Llama-4-Maverick align exactly with our Games domain. On OSWorld, excluding outliers, we observe strong alignment: GPT-4o (1.8%) < Llama-4 (3.0%) ≈ Qwen-32B (3.0%) < Qwen-72B (4.4%), mirroring their AgentVQA standings. Kimi-VL-Instruct is a notable outlier (high OSWorld score vs. low AgentVQA score); however, this model exhibits extreme volatility even across similar offline datasets (ranking 11th on Screenspot but 1st on Screenspot-Pro), suggesting it is an unstable baseline. This is quantitatively confirmed by the Spearman rank correlation, which improves from $\rho \approx 0.143$ to $\rho = 0.700$ when excluding this specific outlier. Overall, these results confirm AgentVQA is a predictive proxy for stable model families.

**Comparison with Original Open-Ended Benchmarks.** We evaluated five representative models on the original open-ended versions of Screenspot, RoboRefit, and MONDAY using their official evaluation protocols. As shown in Table 2, model rankings remain largely consistent within each dataset (e.g., Qwen2.5-VL scaling 32B > 7B > 3B is preserved), confirming that AgentVQA maintains the intrinsic difficulty hierarchy. However, the absolute scores reveal AgentVQA's **calibration effect**: it recovers performance for models penalized by rigid formatting in open-ended settings (e.g., GLM-4V on MONDAY: $15.4\% \to 33.4\%$) while enforcing stricter precision on tasks where original metrics (like bounding box IoU) were too lenient (e.g., Screenspot: $87.1\% \to 73.8\%$).

Table 2: Comparison of accuracy (%) between Original Open-Ended evaluation and AgentVQA (Ours) across three datasets. While rankings are consistent, AgentVQA calibrates scores by penalizing loose grounding (Screenspot) and rescuing valid logic from formatting errors (MONDAY).

| Model | MONDAY | | RoboRefit | | Screenspot | |
|---|---|---|---|---|---|---|
| | Orig | Ours | Orig | Ours | Orig | Ours |
| Qwen2.5-VL-3B | 22.7 | 38.5 | 28.5 | 49.8 | 70.7 | 55.1 |
| Qwen2.5-VL-7B | 36.7 | 44.7 | 44.8 | 54.4 | 80.2 | 59.6 |
| Qwen2.5-VL-32B | 40.6 | 50.6 | 81.5 | 61.1 | 87.1 | 73.8 |
| GLM-4V-Base | 15.4 | 33.4 | 32.4 | 35.0 | 62.4 | 39.3 |
| Gemini-2.5-Pro | 38.4 | 36.7 | 46.5 | 32.8 | 71.4 | 34.8 |

## 4.4 PERFORMANCE ANALYSIS

**Large variance per domain.** Our results in Table 1 suggest that current VLMs are comparatively stronger at tasks involving passive observation and spatial description (Spatial Understanding: 53%, Egocentric Videos: 49%) than at tasks requiring active, goal-directed interaction and planning (Robotics: 40%, Games: 34%, Web Agents: 44%). At the domain level, these patterns hold, but with clear variation across datasets. Within Spatial Understanding and Egocentric Videos, models perform well on EmbSpatial-Bench and Perception-Test (70–80%), whereas others such as VSI-Bench are still challenging.

In contrast, Web Agent datasets like Screenspot and Screenspot-pro remain highly challenging despite scale, while AitW and Mind2Web show relatively higher performance. Robotics have a similar breakdown. While ERQA and Roborefit are tractable (∼60%), Robo2VLM lags behind (50%). Games are universally difficult, with no dataset exceeding 66% even for the strongest reasoning models. These results highlight VLM performance on individual datasets does not necessarily correlate with holistic evaluation, strengthening the call for a unified benchmark like AgentVQA.

Furthermore, our analysis of the 25 distinct sub-task categories defined in Figure 3 (with results in Table 4) suggest that models excel on descriptive, VQA-like tasks such as object counting and entity detection (often >75% for top models). However, they consistently fail on tasks requiring reasoning over future timesteps. The lowest scores across the benchmark are in action prediction (avg. 26%), game-over detection (avg. 27%) and spatial planning (avg. 31%). This highlights a fundamental limitation: current VLMs are proficient at reactive, descriptive tasks but lack the coherent internal world models needed for multi-step planning and strategic reasoning.

**Open-source models are catching up.** There is a clear gap between closed-source and open-source models. The closed-source models score in a range of 40% (GPT-4o) to 60% (GPT-5 thinking-high), while open-source models span a lower range from 35% (Kimi-VL-Instruct) to 53% (Qwen2.5-VL (72B)). While the top closed-source model outperforms the top open-source model, top open-source models are highly competitive in specific domains. For instance, Qwen2.5-VL (72B) always outperforms GPT-4o, and even the small Qwen2.5-VL (3B) model (45%) surpasses GPT-4o (41%) in the Web Agents domain.

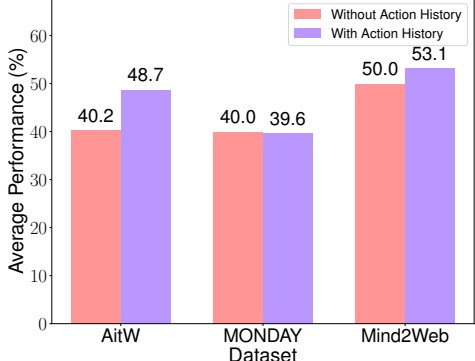

Figure 5: **Impact of action history on model performance in web agent tasks.** The figure shows the average performance change across four representative models (Gemini-2.5-Pro, Llama 4 Scout, Qwen2.5-VL-7B, and Kimi-VL-Thinking) when provided with full trajectory history versus no history.

**Trajectories require action history.** Our ablation on web agent tasks reveals that providing action history is a consistent benefit (Figure 5). Across four models, providing the full trajectory history improved average performance by a substantial 8.5% on AITW and 3.1% on Mind2Web. The performance on MONDAY remained relatively stable (changing by only -0.4%). This confirms that for complex, multi-step tasks, access to historical context is critical for effective decision-making. The stable performance in MONDAY can be hypothesized to the tasks being more state-local, where the immediate visual context is often sufficient for the next action.

**Reasoning offers varying results.** The impact of explicit reasoning is mixed. For the GPT-5 series, the thinking-high variant (avg. 60%) substantially outperforms the thinking-minimal variant (avg. 47%). However, for Kimi-VL and GLM-4.1V, the thinking variants (35% and 43%) offer only marginal overall improvement over instruct or base models (35% and 41%).

This masks a more complex dynamic. In certain domains, the non-reasoning variants actually outperform their reasoning-enabled counterparts (e.g., Kimi-VL in Spatial Understanding (55% for Instruct vs 51% for Thinking) and GLM in Egocentric Videos (56% for Base vs 45% for Thinking)). Our results suggests that for perceptual tasks, base models can be more robust than reasoning models. We found that thinking models, like GLM-4.1V-Thinking and Kimi-VL-Thinking, can get stuck in "thinking loops" – a phenomenon we observed in Appendix K.

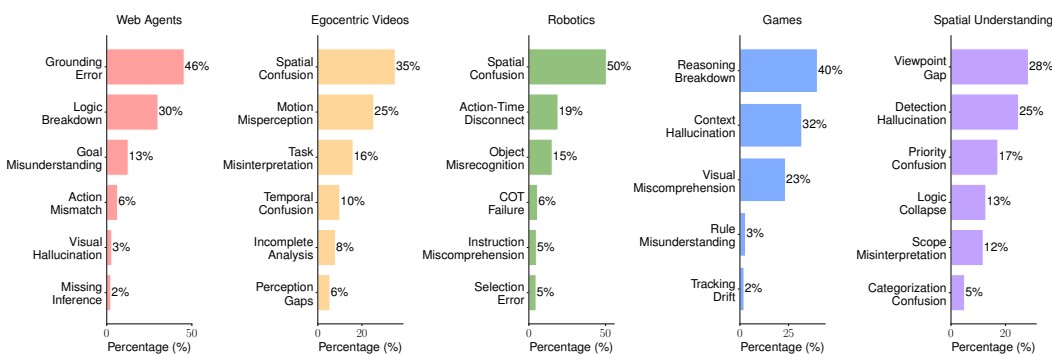

Figure 6: **Distribution of the most common error modes across the five agentic domains in AgentVQA.** The analysis is based on an semi-automated categorization of the incorrect predictions from Gemini-2.5-Pro and Qwen2.5-VL-7B. The chart is organized by domain, with each colored segment representing a primary domain. Each slice within a segment corresponds to a specific error mode, and its size reflects its prevalence (%), summed across the two models.

### 4.5 ERROR MODES

To gain a deeper, qualitative understanding of why models fail, we conduct a comprehensive error mode analysis. We first prompt each model to output its reasoning process along with its final answer.

Then, we subsample 500 error cases per domain and categorize the failures based on these reasoning outputs (approximately evenly distributed per dataset). The outputs are taken from two representative models: a closed-source model (Gemini 2.5 Pro) and an open-source model (Qwen2.5-VL (7B)). To categorize these failures, we first perform a manual analysis to establish a taxonomy of common error modes. Then, we use Gemini to assign each sample to a category. Finally, we manually verified the assignments for accuracy.

Our analysis identifies several recurring failure patterns. The most common error modes vary by domain. The most prevalent error in Web Agents is Grounding Error (46%). In contrast, domains requiring more abstract thought like Robotics and Egocentric Videos are dominated by various forms of spatial confusion (51% and 35% respectively). The spatial understanding domain is dominated by viewpoint gap (28%). Finally, in the games domain, the most frequent issue is reasoning breakdown (40%), meaning that the VLM incorrectly reasoned about the game mechanics or the agent dynamics.

The distribution of these errors, shown in Figures 10 and 6, reveals that failure modes are domain *and* model specific. In Web Agents, grounding error is far more pronounced in Gemini (50%). Conversely, in Games, while reasoning breakdown is a common error for Qwen, Gemini suffers from it to a much lesser extent. This suggests that different models have distinct architectural failure points; some are more prone to failing at the initial perception and grounding step, while others are more likely to perceive correctly but error in subsequent logical steps. More details about the error modes and their descriptions is available in Appendix H.

## 5 CONCLUSION

We introduce AgentVQA, a unified benchmark for evaluating VLM agents. Unlike existing benchmarks that are non-agentic, focus on traditional VQA, or are domain-specific, AgentVQA consolidates agentic benchmarks into a balanced MCQ dataset with hard negatives. Evaluation of 15 state-of-the-art VLMs shows a significant performance gap, with top models achieving only ∼60% accuracy. VLM rankings on AgentVQA differ substantially from traditional benchmarks, demonstrating AgentVQA's value as a challenging evaluation framework for generalizable, agentic VLMs.

**LLM Usage Statement.** We used LLMs to help automate aspects to the manual review and summarization of our datasets. Additionally, we used LLMs to help improve sentence quality and improve conciseness.

**Reproducibility Statement.** To ensure our research is transparent and reproducible, all components of the AgentVQA benchmark will be made publicly available upon publication. This includes the full dataset, evaluation code, and detailed results. Our dataset is constructed from 14 publicly available sources (detailed in Section 3.2 and Appendix A), and the scripts for our VLM-assisted data generation pipeline will also be released. All models evaluated are either open-source or were accessed via standard APIs, with specifics provided in our paper. The complete evaluation codebase and configurations will be released to allow for the precise replication of our findings.

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

## A  DATASET SELECTION AND CURATION RATIONALE

The construction of AgentVQA is guided by a principled selection process to ensure the final benchmark is comprehensive, robust, and effective for evaluating the agentic capabilities of Vision-Language Models (VLMs). Our domains: Web Agents, Egocentric Videos, Robotics, Games, and Spatial Understanding are chosen to represent a broad spectrum of agentic challenges, from digital interaction to embodied physical perception and strategic planning. The following five criteria are central to our selection of the 14 source datasets.

**High agentic signal**  A primary requirement is that each dataset's tasks must involve a strong "agentic signal." This means moving beyond passive visual question answering (e.g., "What color is the car?") to scenarios that demand active reasoning from an agent's perspective. We prioritize datasets where the core task involves:

- Choosing actions or predicting the next optimal step.
- Evaluating the consequences of a sequence of actions.
- Understanding causality and the relationships between actions and outcomes.
- Strategic planning toward a specific goal.

This focus ensures that AgentVQA directly probes the decision-making and reasoning capabilities that are fundamental to agentic intelligence.

**Comprehensive and distinct domain coverage**   To test for generalist agentic behavior, we select datasets that span diverse domains and modalities. Each of the five chosen domains introduces a unique set of challenges and required skills not fully covered by the others:

- Web Agents: Tests GUI navigation, grounding, and interaction in a digital environment.
- Robotics & Egocentric Videos: Evaluate embodied perception, spatial awareness, and understanding of the physical world from a first-person perspective.
- Games: Focus on strategic decision-making, reward modeling, and long-term planning in rule-based environments.
- Spatial Understanding: Assesses foundational spatial awareness and reasoning, a critical component for any embodied agent.

This multi-domain approach prevents the benchmark from being too narrow and allows for a holistic assessment of a VLM's generalization ability.

**Offline convertibility**   A crucial practical consideration is the feasibility of converting each dataset into a standardized, offline format. This criterion requires that the source data (e.g., trajectories, videos) be sufficiently structured to allow for an unambiguous conversion into a Multiple-Choice Question (MCQ) format. Every resulting MCQ must have a single, verifiably correct answer to enable scalable, deterministic, and efficient evaluation without relying on resource-intensive online simulators.

**Public availability and permissive licensing**   To ensure transparency, reproducibility, and accessibility for the broader research community, all selected datasets must be publicly available. Furthermore, their licenses must permit modification and redistribution as part of a new benchmark, allowing us to legally and ethically create and share AgentVQA.

**Sufficient scale**   The selected datasets need to be large enough to allow for statistically meaningful evaluation. We generally subsample 1,000 instances from each source dataset. Our analysis confirms that this sample size consistently yields a standard deviation in model performance of less than 1%, ensuring our results are stable and reliable. An exception is made for the ERQA dataset, from which we curate 400 instances. Despite its smaller size, its exceptional quality and high relevance to core robotic reasoning challenges make its inclusion essential.

## B    Hard Negative Generation Pipeline

The difficulty of AgentVQA is ensured by a systematic process for generating challenging distractor options, or "hard negatives." This process transforms tasks that are not inherently multiple-choice, such as web navigation trajectories, into a standardized and rigorous MCQ format. Our pipeline creates negatives that are both plausible and unambiguously incorrect, forcing models to demonstrate fine-grained understanding rather than relying on simple heuristics. The process is detailed below.

**VLM-assisted generation.**   To select the generation model, we first conducted a preliminary study on a data subset using Gemini 2.5 Pro, GPT-4o, and Qwen 2.5 VL (7B). After manually comparing the generated samples for both difficulty and accuracy, we selected Gemini 2.5 Pro for its superior performance. We therefore employ a VLM-assisted pipeline centered around **Gemini 2.5 Pro** to generate hard negatives. For each question, we provide the model with comprehensive context, including the primary image or screenshot, the task prompt, the ground-truth answer, any relevant action history for sequential tasks, and metadata such as image dimensions and coordinate formats (e.g., (x,y) for points, or $(x_1, y_1, x_2, y_2)$ for bounding boxes). For grounding tasks, we also visually render the correct action's location onto the image. Given this context, we prompt the model to generate three distinct hard negatives as a combination of two types: **near-miss negatives** and **semantically similar negatives**. Near-miss negatives are distractors spatially close to the correct answer that test precision, such as coordinates a few pixels away from a correct UI element.

Semantically similar negatives are conceptually related to the correct answer and test the model's ability to differentiate between similar objects, such as choosing a "Mobile" phone when the correct answer is a "Tablet." The exact prompt used for generating these "hard negatives" is in Appendix M

**Verification and filtering.** To ensure quality, the generated negatives undergo a two-stage verification process. First, an automated script filters grounding-based distractors by verifying their coordinates fall outside the ground-truth bounding box, programmatically confirming the option is a "negative." Following this, a subset of these samples undergoes a rigorous manual review. We manually verify each distractor for both plausibility, ensuring the option is a believable choice that could confuse a model, and unambiguous incorrectness, which guarantees there is only one correct answer to the MCQ. This final human-in-the-loop step is critical for guaranteeing the integrity and challenge of the AgentVQA benchmark.

## C DEFINITION OF AGENTIC TASKS

The community defines agentic tasks as those requiring a system to "observe their environment and act in it in order to achieve goals" (Bengio et al., 2025), encompassing perception to sense the state, intelligence to reason and plan, and affordances to execute actions (Plaat et al., 2025). While traditional image VQA treats visual understanding as an end in itself, agentic VQA frames perception as a means to action. Here, each answer must inform executable decisions rather than merely describe what is seen (Reed et al., 2022). To ensure AgentVQA captures this full capability range, we structure our 25 sub-tasks into a unified hierarchy: Action Grounding tasks (e.g., Tap) test the critical translation of semantic intent into executable spatial actions (Reed et al., 2022); Spatiotemporal Reasoning tasks evaluate the dynamic mental maps required for movement; State Understanding tasks (e.g., Function Reasoning) ensure the agent can accurately parse the environment to inform decision-making; and Strategic Planning tasks (e.g., Reward Modeling) assessing long-term outcomes through multi-step reasoning (Sutton et al., 1998).

## D SUB-TASK CATEGORIZATION METHODOLOGY

To enable a granular analysis of model capabilities, we structure the questions in AgentVQA into 25 distinct sub-task categories. This provides a fine-grained breakdown of the specific agentic skills under evaluation. The creation and assignment of these categories follows a systematic methodology, beginning with manual taxonomy design and domain-specific programmatic mapping. The process begins with a manual inspection of a representative subset of the benchmark, ensuring equal distribution across all domains and datasets. From this analysis, we established a new, unified taxonomy of 25 sub-task categories that best represent agentic skills. Following this, we programmatically mapped every question to one of these categories, with logic tailored to each domain.

**Domain-specific mapping details.** For the Spatial Understanding domain, we adopted the category structure from SpaCE-10 and mapped questions from EmbSpatial-Bench by using pre-existing metadata. Questions involving the relations 'above', 'under', 'left of', 'right of', 'on', 'in', 'behind', 'in front of', or 'touching' were mapped to our Object-Object Interaction category. Questions with 'close' or 'far' relations were mapped to Object-Scene Interaction. For Web Agents, we extracted action keywords from the ground-truth text, except for datasets like Screenspot where all actions defaulted to Tap. For Robotics, we mapped fine-grained, pre-existing categories from source datasets to our unified high-level categories. For example, in ERQA, 'Action Reasoning' and 'Trajectory Reasoning' were mapped to Task Estimation, while in Robo2VLM, 'relative_depth' and 'view_correspondence' were mapped to POV Modeling. For Egocentric Videos, we used a similar approach; in VSI-Bench, categories like 'route_planning' were mapped to Spatial Navigation, while in Perception-Test, questions were categorized using keywords, such as those starting with "how many" being mapped to Object Counting. Finally, for Games, existing categories from GameQA were mapped to the bespoke categories we designed for the Atari dataset; for instance, 'target perception' and 'state prediction' were mapped to World Modeling, and 'strategy optimization' was mapped to Reward Modeling.

### D.1 Sub-Task Category Definitions

**Web agents.** This domain tests interaction with graphical user interfaces. Tap and Press involve grounding a click or selection action on the correct UI element. Scroll tests the ability to infer the need for vertical or horizontal movement to reveal off-screen elements. Typing involves inputting text into the correct field.

**Egocentric videos.** This domain focuses on understanding first-person video. Object Counting requires quantifying static items in a scene, testing core visual perception. Spatial Navigation tests the ability to build a mental map of an environment and understand orientation from a specific viewpoint. Motion Causality involves interpreting dynamic events, understanding physical interactions, and reasoning about the consequences of actions over time.

**Robotics.** This domain assesses reasoning in physical, embodied scenarios. Object Grounding involves correctly identifying a target object based on its unique attributes, especially when similar distractors are present. POV Modeling requires understanding 3D spatial relationships, orientation, and object positions from a 2D camera perspective. Task Estimation tests the comprehension of a task's overall goal and the logical sequence of actions required to achieve it. State Tracking involves assessing the current status of an object or the environment, such as its stability or configuration.

**Games.** This domain evaluates strategic and predictive reasoning in rule-based environments. Action Prediction involves determining the next valid move. Game-Over Detection requires recognizing conditions that end the game. Reward Detection involves identifying specific events that trigger a score change. Reward Modeling requires evaluating states or actions to determine which leads to a better outcome. Game Strategization involves high-level, multi-step planning. World Modeling tests the ability to predict a future game state given a sequence of actions.

**Spatial understanding** This domain tests foundational spatial intelligence. Entity Detection and Entity Counting involve identifying the presence and number of objects. Object Sizing and Scene Counting assess attributes of objects and the environment. Object-Object Interaction and Object-Scene Interaction require reasoning about the relative spatial relationships between different elements. Functional Reasoning and Spatial Planning test a deeper understanding of how objects can be used and how an agent can navigate or manipulate the environment to achieve a goal.

## E   Subsampling Robustness Analysis

To ensure AgentVQA is both manageable and statistically reliable, we conducted a robustness analysis to determine the optimal number of instances to subsample from each source dataset. Our goal was to find the smallest sample size that provides a stable and accurate estimate of model performance compared to the full dataset. We defined our stability criterion as achieving a 95% confidence interval width of less than 1.0% absolute.

**Methodology.** Our analysis begins by first running a full evaluation on the entire dataset to obtain a ground-truth response (correct or incorrect) for every question. Using this complete set of responses, we efficiently simulate the subsampling process by drawing smaller subsets of varying sizes to observe how the average accuracy changes. To dig deeper into the variance, we focus on subset sizes of 300, 500, 700, 1000, and 1200. For each of these sizes, we draw 50 independent, stratified random samples from our saved full-dataset results. Stratified sampling ensures that the proportional representation of sub-task categories is maintained. For each of the 50 runs, we compute the mean accuracy, sample standard deviation, and the standard error of the mean. Using a t-value for a 95% confidence level with 49 degrees of freedom, we construct the confidence interval for the mean accuracy at each sample size.

**Results.** The analysis, summarized in Figures 7, 8 and 9, demonstrate that as the sample size increases, the mean accuracy quickly converges to the full dataset's performance, and the variance across runs decreases significantly. Our results show that a sample size of 1000 is the smallest size that consistently meets our stability criterion, yielding a confidence interval width of 0.75%. We observe

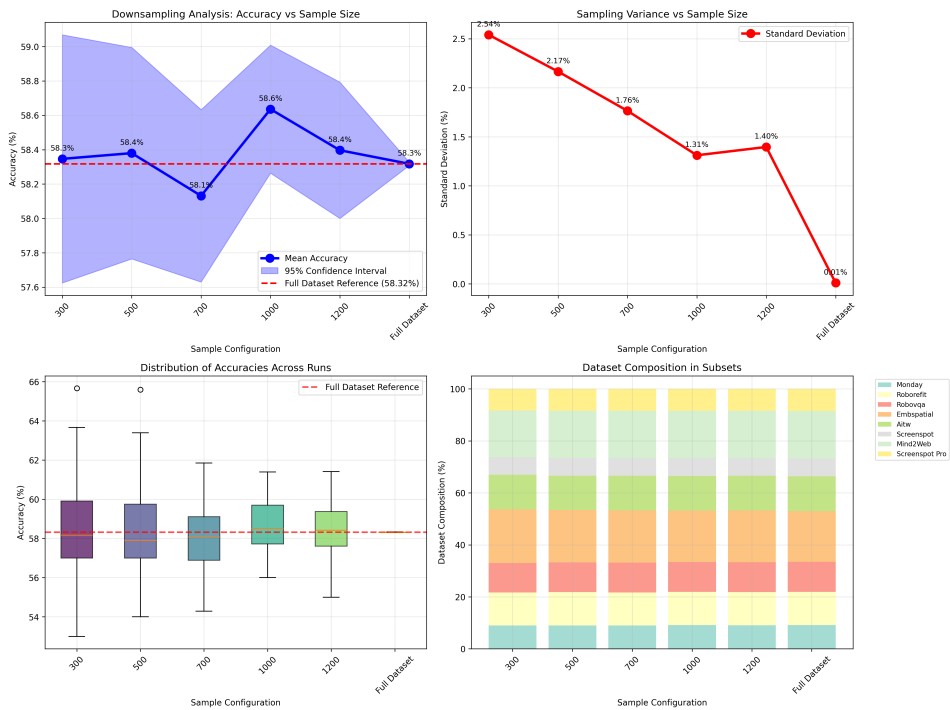

Figure 7: **Overall subsampling analysis.** The plots show that as the sample size increases towards the full dataset size, the mean accuracy (top-left) stabilizes and the sampling variance (top-right) decreases, demonstrating the reliability of using a large subsample.

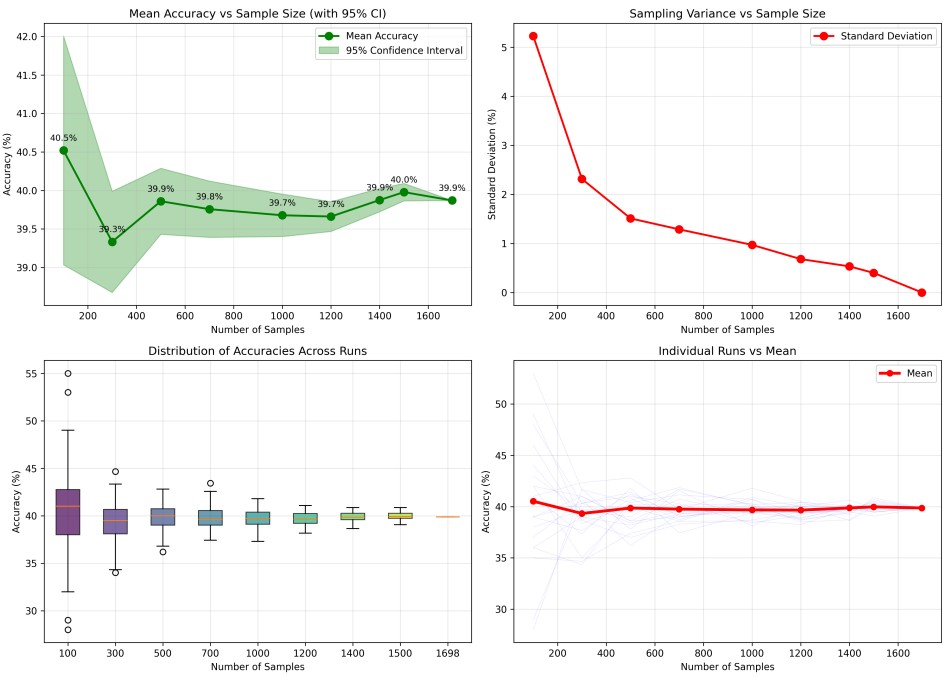

Figure 8: **Detailed robustness analysis for the MONDAY dataset.** This view illustrates how the 95% confidence interval (top-left, green shade) narrows significantly as the sample size increases, providing a stable estimate of the mean accuracy.

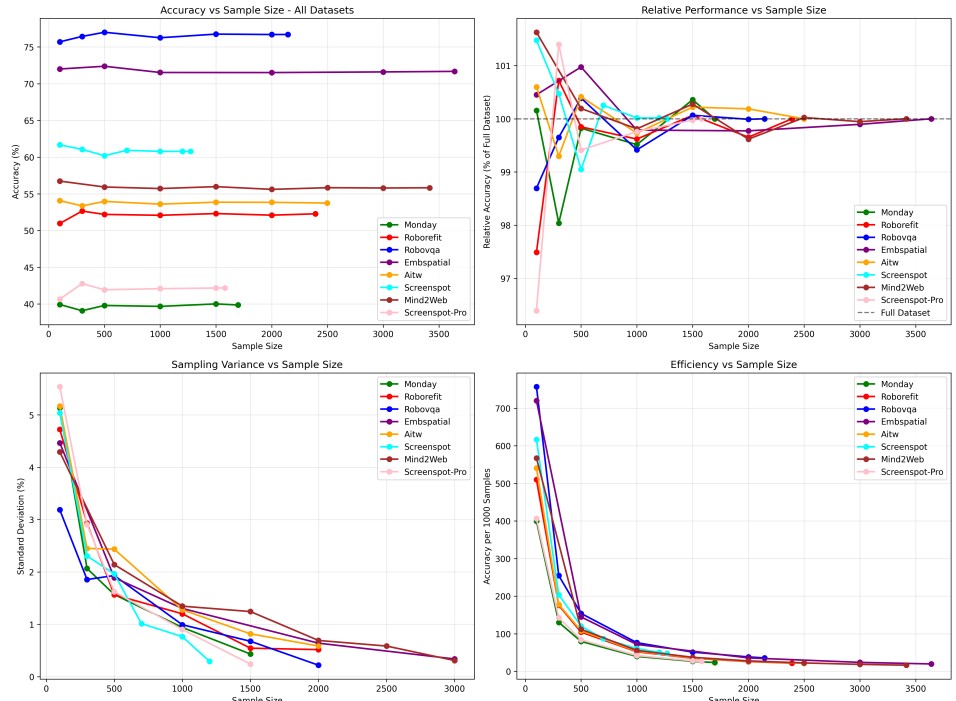

Figure 9: **Comparative subsampling analysis across multiple datasets.** The bottom-left plot is particularly illustrative, showing a consistent trend of decreasing sampling variance (standard deviation) for all datasets as the sample size approaches 1000.

that the 1000-sample size provides a strong balance of efficiency and statistical confidence. This empirical validation justifies our decision to standardize the benchmark's datasets to 1000 instances each, ensuring that the performance metrics are reliable indicators of a model's true capabilities.

## F    ANSWER EXTRACTION LOGIC

To ensure consistent and automated evaluation, we employ a multi-strategy answer extraction logic. Although the prompt instructs models to output only a single letter corresponding to their final choice, outputs can vary. Our programmatic parser is designed to robustly handle these variations.

**Extraction strategy.**    The extraction process follows a prioritized sequence of rules. First, our parser attempts a direct match, checking if the model's entire stripped output is exactly a single, valid option letter (e.g., 'A'). If this primary strategy fails, it deploys a fallback routine that applies a series of regular expression patterns. These patterns are designed to find common answer declarations such as "Answer: A," or "Option A." We note that these regular expressions are occasionally adapted to accommodate the unique, consistent output formats of specific models. If no unambiguous answer can be identified after applying all strategies, the question is skipped and marked unevaluated to maintain the integrity of the evaluation. The complete prompt is available in Appendix L.

## G    ROBUSTNESS ANALYSIS OF MCQ FORMAT

To ensure the reliability of the multiple-choice format, we conducted extensive ablation studies to quantify the impact of option ordering and rule out random guessing as a confounding factor.

**Robustness to Option Ordering.**    We evaluated 5 representative models across 6 datasets using 5 distinct, seeded random permutations of option orders. As shown in Table 3, the model performance

remains highly consistent across permutations, with a standard deviation of approximately 1.9%. This confirms that the rankings reported in AgentVQA are stable artifacts of model capability rather than positional bias.

Table 3: Robustness to Option Ordering: Average accuracy across 5 random permutations of option orders. The low standard deviation indicates high stability.

| Shuffle Iteration | 0 | 1 | 2 | 3 | 4 | Std. Dev |
|---|---|---|---|---|---|---|
| Average Accuracy | 56.0% | 54.8% | 57.0% | 51.7% | 54.5% | ~ 1.9% |

**Refuting Random Guessing.** To verify that model scores reflect genuine engagement with the task, we evaluated constant-selection baselines (e.g., "Select All A", "Select All B") across 5 datasets. These strategies yielded an average accuracy of **24.9%** (ranging from 18.6% to 28.9% across different options), which aligns with the theoretical random baseline for 4-option MCQs. Since even the lowest-performing models in AgentVQA consistently score significantly above this floor, we conclude that the benchmark effectively measures agentic reasoning rather than chance.

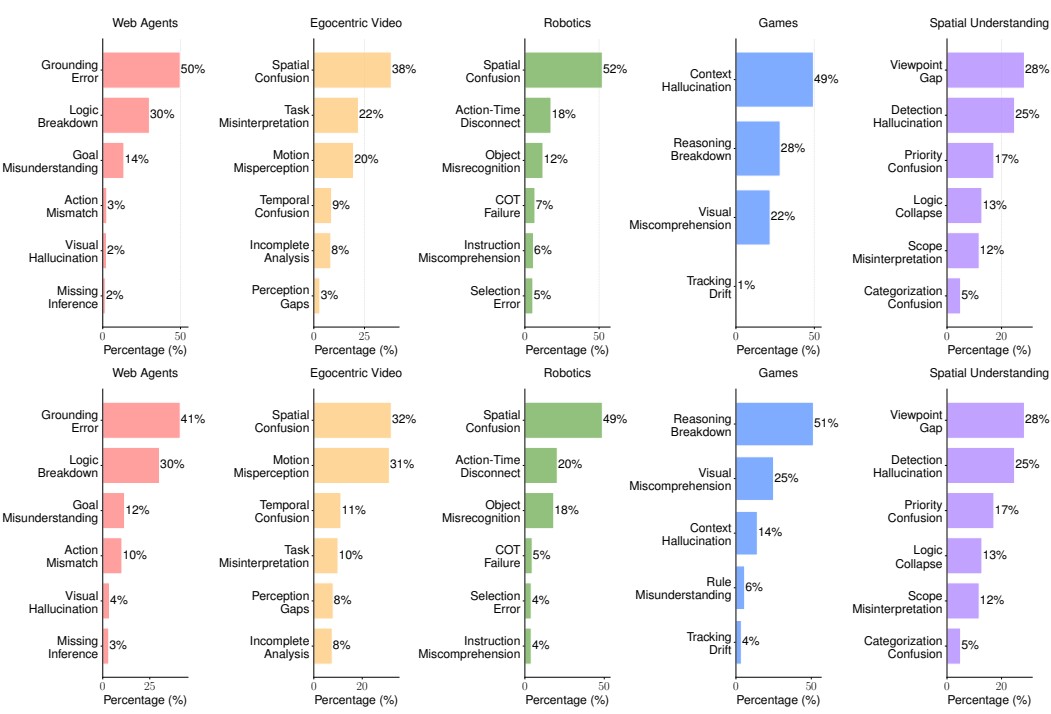

Figure 10: **Distribution of the most common error modes across the five agentic domains in AgentVQA.** The analysis is based on a semi-automated categorization of incorrect predictions from Gemini 2.5 Pro (top) and Qwen-2.5VL (7B) (bottom). In both charts, each colored segment represents a primary domain, and each slice within a segment corresponds to a specific error mode, with its size reflecting its prevalence (%).

## H   ERROR MODES

To qualitatively understand model failures, we conduct an in-depth error mode analysis on two representative models: a closed-source model (Gemini 2.5 Pro) and an open-source model (Qwen2.5-VL (7B)). The process involves a three-stage methodology to develop a taxonomy, annotate failures, and verify the results. The resulting distribution of errors is visualized in Figure 10.

**Taxonomy development and annotation**    First, we prompt the models to output a reasoning chain alongside their final answer for every question. We then manually review a diverse, stratified subset of incorrect predictions to identify and define a comprehensive taxonomy of common failure patterns for each of the five agentic domains. With this taxonomy established, we use Gemini 2.5 Pro for large-scale annotation. For each failure, we provide the model with the visual input, all multiple-choice options, the ground-truth answer, the model's incorrect chosen option, the model's full reasoning chain, and any other relevant metadata. We then prompt it to assign the single most fitting error mode from our predefined taxonomy. The prompt used for assigning error modes using Gemini is available in Appendix N.

**Verification**    To ensure the quality of the automated annotations, we perform a final verification step. We manually review a randomly selected subset of the error mode assignments across all domains and for both models. This process confirms that the classifications made by Gemini 2.5 Pro are accurate and consistent with our established definitions, ensuring the reliability of our qualitative analysis.

## H.1   ERROR MODE DEFINITIONS

**Web agents.**    Grounding Error occurs when the model understands the goal but fails to correctly locate or ground the corresponding UI element, often confusing it with a visually similar or nearby one. Logic Breakdown describes a failure to correctly process a sequence of steps, leading to a logically inconsistent action based on the current state. Goal Misunderstanding is a fundamental error where the model misinterprets or ignores a key part of the user's instruction. Action Mismatch happens when the model's reasoning correctly identifies the right action, but its final output selects a different, incorrect option. Missing Inference occurs when the model fails to deduce an implicit but necessary action, such as scrolling to reveal an element. Visual Hallucination is when the model's reasoning relies on a UI element or piece of information that is not present in the visual input.

**Egocentric video.**    Spatial Confusion is a failure to build an accurate 3D mental map from video, resulting in an incorrect understanding of object layout or orientation from a given viewpoint. Motion Misperception occurs when the model fails to correctly interpret a process, motion, or physical interaction as it unfolds over time. Task Misinterpretation arises when visual perception is correct, but the model misunderstands the nuance or goal of the prompt. Temporal Confusion is a sequencing error where the model correctly perceives individual events but gets their chronological order wrong. Incomplete Analysis happens when the model perceives all necessary visual facts but fails to execute the required logical steps to reach the correct conclusion. Perception Gaps are fundamental failures in seeing, where the model incorrectly reports a basic, static fact like an object's presence or count.

**Robotics.**    Spatial Confusion occurs when the model fails to correctly interpret the spatial relationships, location, or orientation of objects in 3D space from a 2D image. Action-Time Disconnect is a failure to understand the dynamics of a scene, such as predicting the outcome of an action or correctly sequencing a task. Object Misrecognition happens when the model misidentifies an object or fails to ground it based on all of its required attributes like color or size. Instruction Miscomprehension occurs when the model perceives the scene correctly but fundamentally misunderstands the user's goal. COT Reasoning Gaps describe failures to handle specific quantitative values or fine-grained distinctions within the reasoning process. Output Mapping Error is when the model's reasoning is correct, but it outputs the wrong corresponding letter for its final answer.

**Games.**    Reasoning Breakdown is a general failure where the model's internal logic collapses under the task's complexity or becomes confused about the objective, leading to a flawed conclusion. Context Hallucination occurs when the model's reasoning becomes detached from the game state, inventing information that does not exist. Visual Miscomprehension is a foundational failure to correctly perceive the game's visual and spatial information, such as object locations or relationships. Rule Misunderstanding is a logical error where the model does not correctly apply the game's explicit rules or objectives. State Tracking Drift is a failure to mentally simulate and track how the game state changes over time.

**Spatial understanding.**    Viewpoint gaps occur when the model identifies objects but fails to construct an accurate 3D map of their relationships, leading to errors in judging depth or distance.

Detection Hallucination is a fundamental perceptual failure where the model either misses a present object or "sees" an object that is not there. Priority Confusion is a subtle reasoning error where the model correctly identifies multiple true spatial facts but incorrectly prioritizes one that is irrelevant to the question. Logic Collapse is a failure cascade where the model first fails to perceive an object and then tries to compensate by inventing a logical path forward based on hallucinated information. Scope Misinterpretation happens when the model misunderstands the required granularity or intent of the question, particularly for counting or classification tasks. Categorization Confusion occurs when perception is visually correct, but the model applies the wrong semantic label or category to an object.

Table 4: Comprehensive Evaluation of 15 VLMs on AgentVQA Across 5 Agentic Domains. 🤔 represents Reasoning Model. 🔒 represents closed-source models.

| Model | Web Agents | | | | | Robotics | | | | | Egocentric-Videos | | | |
|---|---|---|---|---|---|---|---|---|---|---|---|---|---|---|
| | Tap | Typing | Scroll | Press | Avg. | TE | POV | ST | OG | Avg. | SN | MC | OC | Avg. |
| GPT 5 thinking-high 🤔🔒 | 55.3 | 74.4 | 47.6 | 37.1 | 55.2 | 53.9 | 38.5 | 63.9 | 52.1 | 51.2 | 63.5 | 68.8 | 76.0 | 70.4 |
| Gemini 2.5 Pro 🔒 | 49.3 | 72.5 | 49.2 | 41.9 | 50.3 | 47.9 | 34.3 | 28.3 | 39.5 | 40.2 | 40.3 | 48.3 | 67.4 | 54.4 |
| GPT 5 thinking-minimal 🔒 | 44.3 | 71.4 | 38.9 | 46.3 | 45.6 | 47.7 | 36.8 | 49.7 | 36.0 | 43.7 | 58.4 | 49.6 | 84.3 | 68.0 |
| GPT 4o 🔒 | 40.9 | 65.0 | 20.6 | 38.3 | 40.5 | 42.6 | 32.7 | 44.6 | 19.3 | 35.4 | 30.8 | 53.8 | 50.0 | 44.9 |
| GLM-4.1V-Thinking 🤔 | 40.9 | 71.1 | 36.2 | 35.9 | 42.3 | 38.7 | 41.0 | 28.9 | 24.8 | 34.6 | 29.7 | 37.7 | 58.5 | 44.6 |
| Kimi-VL-Thinking 🤔 | 32.0 | 58.6 | 33.1 | 34.6 | 33.1 | 38.4 | 34.2 | 31.3 | 36.0 | 36.9 | 41.0 | 22.5 | 23.9 | 29.0 |
| Phi-4 Multimodal | 30.0 | 63.3 | 48.2 | 44.7 | 33.7 | 25.9 | 32.1 | 43.1 | 37.1 | 32.6 | 45.5 | 38.6 | 72.1 | 55.9 |
| GLM-4.1V-Base | 37.5 | 69.8 | 41.8 | 53.4 | 40.3 | 36.3 | 33.1 | 36.7 | 21.9 | 32.4 | 15.0 | 30.9 | 41.0 | 30.5 |
| Kimi-VL-Instruct | 32.8 | 40.5 | 24.3 | 19.1 | 34.3 | 38.1 | 30.2 | 26.0 | 35.4 | 37.4 | 41.6 | 38.1 | 52.9 | 45.9 |
| Llama-4-Scout | 42.2 | 65.4 | 37.2 | 25.5 | 42.1 | 29.9 | 18.9 | 50.8 | 30.6 | 34.7 | 40.8 | 35.4 | 53.3 | 45.2 |
| Llama-4-Maverick | 37.9 | 66.3 | 41.7 | 27.9 | 39.6 | 36.8 | 18.8 | 37.2 | 36.0 | 35.8 | 38.3 | 36.0 | 55.8 | 45.7 |
| Qwen 2.5 VL (3B) | 44.6 | 64.3 | 34.9 | 35.9 | 44.6 | 32.6 | 30.2 | 43.8 | 52.4 | 37.9 | 35.8 | 37.8 | 65.5 | 49.7 |
| Qwen 2.5 VL (7B) | 50.6 | 71.8 | 42.1 | 46.2 | 51.5 | 40.4 | 35.3 | 57.8 | 55.6 | 44.4 | 34.0 | 41.5 | 69.1 | 51.6 |
| Qwen 2.5 VL (32B) | 54.8 | 74.5 | 46.3 | 52.2 | 55.0 | 43.7 | 34.2 | 58.3 | 62.0 | 47.7 | 39.6 | 36.7 | 67.3 | 51.4 |
| Qwen 2.5 VL (72B) | 57.4 | 77.3 | 50.9 | 52.5 | 57.3 | 48.3 | 39.9 | 57.1 | 61.4 | 51.5 | 33.0 | 43.7 | 68.5 | 51.5 |

| Model | Games | | | | | | | Spatial Understanding | | | | | | | | |
|---|---|---|---|---|---|---|---|---|---|---|---|---|---|---|---|---|
| | WM | RM | GS | AP | GOD | RD | Avg. | ED | OO | EC | FR | OSI | OS | SP | SC | Avg. |
| GPT 5 thinking-high 🤔🔒 | 60.0 | 94.0 | 67.5 | 31.9 | 51.4 | 36.9 | 60.2 | 83.6 | 84.0 | 31.3 | 45.4 | 60.9 | 82.7 | 57.9 | 46.8 | 71.5 |
| Gemini 2.5 Pro 🔒 | 33.4 | 50.5 | 27.5 | 34.5 | 34.5 | 37.5 | 39.5 | 31.3 | 60.9 | 25.1 | 45.8 | 54.7 | 65.3 | 37.5 | 31.4 | 59.5 |
| GPT 5 thinking-minimal 🔒 | 42.4 | 84.5 | 31.1 | 30.5 | 31.5 | 40.0 | 47.4 | 41.8 | 71.4 | 33.3 | 58.0 | 55.4 | 70.4 | 26.8 | 48.0 | 66.5 |
| GPT 4o 🔒 | 49.3 | 21.7 | 43.7 | 26.5 | 18.6 | 20.7 | 31.4 | 77.3 | 76.3 | 28.9 | 43.6 | 54.1 | 78.2 | 44.4 | 35.4 | 51.1 |
| GLM-4.1V-Thinking 🤔 | 39.1 | 30.3 | 25.5 | 25.7 | 26.3 | 29.0 | 35.2 | 57.1 | 79.4 | 29.8 | 50.0 | 55.0 | 70.7 | 36.8 | 24.4 | 62.2 |
| Kimi-VL-Thinking 🤔 | 25.7 | 24.9 | 26.9 | 23.7 | 31.5 | 26.5 | 24.0 | 30.7 | 75.8 | 22.4 | 32.6 | 43.5 | 34.2 | 27.5 | 31.3 | 50.8 |
| Phi-4 Multimodal | 25.9 | 58.6 | 22.5 | 27.1 | 24.6 | 25.6 | 26.6 | 46.8 | 67.9 | 33.4 | 47.0 | 46.5 | 53.0 | 37.2 | 15.9 | 58.0 |
| GLM-4.1V-Base | 26.6 | 35.3 | 20.4 | 18.0 | 27.2 | 19.9 | 23.2 | 28.5 | 69.9 | 30.3 | 42.2 | 54.1 | 62.1 | 35.0 | 51.4 | 55.0 |
| Kimi-VL-A3B-Instruct | 25.4 | 22.8 | 23.6 | 31.5 | 23.0 | 22.5 | 28.1 | 25.1 | 88.9 | 30.9 | 39.3 | 44.6 | 51.0 | 24.7 | 27.4 | 51.1 |
| Llama-4-Scout | 29.5 | 62.0 | 23.1 | 23.0 | 28.5 | 30.0 | 32.4 | 29.3 | 62.1 | 34.0 | 39.7 | 47.0 | 52.6 | 26.3 | 54.5 | 49.2 |
| Llama-4-Maverick | 29.1 | 78.5 | 19.8 | 19.0 | 15.5 | 31.0 | 28.5 | 35.2 | 57.1 | 31.2 | 56.9 | 47.6 | 49.3 | 5.7 | 45.7 | 52.3 |
| Qwen 2.5 VL (3B) | 24.0 | 28.0 | 27.7 | 23.5 | 29.0 | 26.3 | 24.8 | 16.8 | 64.0 | 20.2 | 37.0 | 42.7 | 47.4 | 21.1 | 12.8 | 45.2 |
| Qwen 2.5 VL (7B) | 30.4 | 28.7 | 30.2 | 24.2 | 17.7 | 30.7 | 29.1 | 26.6 | 72.2 | 31.6 | 41.3 | 54.2 | 51.4 | 20.6 | 39.3 | 55.3 |
| Qwen 2.5 VL (32B) | 33.3 | 49.7 | 33.2 | 24.7 | 20.7 | 26.2 | 32.6 | 27.2 | 78.6 | 37.2 | 51.4 | 52.7 | 64.7 | 31.6 | 41.0 | 60.1 |
| Qwen 2.5 VL (72B) | 40.8 | 73.1 | 34.8 | 32.1 | 23.6 | 31.6 | 40.5 | 47.8 | 76.1 | 35.1 | 59.6 | 52.7 | 60.9 | 26.3 | 43.0 | 61.0 |

Table 5: Here we list canonical Vision-Language model agentic benchmarks, organized by domain and their offline/online categorization. We **bold** datasets that are apart of AgentVQA and suffix datasets that have been filtered or filtered and transformed with + or ++ suffixes, respectively.

| Domain | Mode | Datasets |
|---|---|---|
| Web Agents | Offline | **AitW**++ (Rawles et al., 2023), **Mind2Web**++ (Deng et al., 2023), **ScreenSpot**++ (Cheng et al., 2024a), **Monday**++ (Jang et al., 2025), **Screenspot-Pro**++ (Li et al., 2025) |
| | Online | WebArena (Zhou et al., 2023), VisualWebArena (Koh et al., 2024), TheAgentCompany (Xu et al., 2024), AndroidWorld (Rawles et al., 2024), VideoWebArena (Jang et al., 2024) |
| Egocentric Videos | Offline | **Perception-Test**+ (Patraucean et al., 2023), **VSI-Bench**+ (Yang et al., 2025a), VidEgoThink (Cheng et al., 2024b), Ego-Exo4D (Grauman et al., 2024), OpenEQA (Majumdar et al., 2024) |
| | Online | EgoThink (Cheng et al., 2024c), EgoPlan-Bench (Chen et al., 2023) |
| Robotics | Offline | **RoboRefit**++ (Lu et al., 2023), **ERQA** (Team et al., 2025), **Robo2VLM**+ (Chen et al., 2025), HoloAssist (Wang et al., 2023), X-Embodiment (Vuong et al., 2023) |
| | Online | CALVIN (Mees et al., 2022), VIMA-Bench (Jiang et al., 2022), BEHAVIOR-1K (Li et al., 2023b), RoboCasa (Nasiriany et al., 2024), EmbodiedBench (Yang et al., 2025b) |
| Games | Offline | **Atari**++ (Zhang et al., 2020), **GameQA**+ (Tong et al., 2025), MarioQA (Mun et al., 2017), BASALT (Milani et al., 2023) |
| | Online | Crafter (Hafner, 2021), MineDojo (Fan et al., 2022), VideoGameBench (Zhang et al., 2025), BALROG (Paglieri et al., 2024) |
| Spatial Understanding | Offline | **EmbSpatial-Bench**+ (Du et al., 2024), **SpaCE-10**+ (Gong et al., 2025), Spatial-MM (Shiri et al.), OmniSpatial (Jia et al., 2025), 3DSRBench (Ma et al., 2024) |
| | Online | HabitatChallenge (Savva et al., 2019), EXCALIBUR (Zhu et al., 2023) |

# I   LIMITATIONS OF OPEN-ENDED GROUNDING METRICS

In this section, we visualize why standard open-ended evaluation metrics (Bounding Box IoU and Distance Thresholds) introduce significant noise into agentic evaluation, and how AgentVQA's MCQ format resolves this ambiguity. As illustrated in Figure 11, open-ended metrics often force a

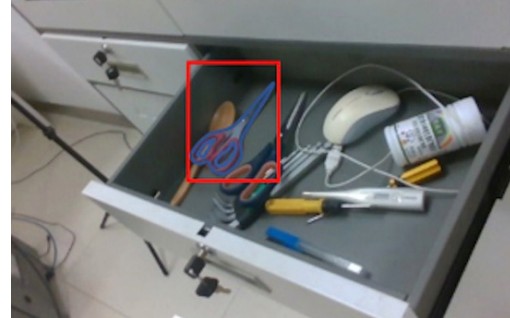 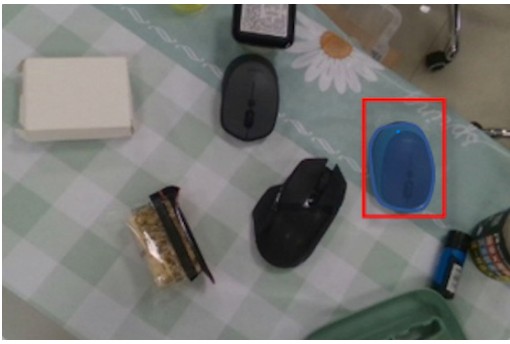

(a) Ambiguous Bounding Box (Scissors)          (b) Ambiguous Bounding Box (Mouse)

Figure 11: **Failure Modes of Open-Ended Grounding Evaluation. (a) False Positive Risk:** The target object (scissors) has a highly irregular, diagonal shape. A standard rectangular bounding box (red) necessarily includes a large area of empty space and adjacent objects (e.g., the wooden spoon). A model prediction falling into this empty void would be counted as a "hit" under standard IoU metrics despite missing the object itself. **(b) Threshold Sensitivity:** For the dark blue wireless mouse, the bounding box is relatively large. A strict center-point distance threshold might reject a valid click near the edge of the mouse, while a loose threshold could incorrectly credit a click on the adjacent black mouse.

trade-off: loose thresholds or large bounding boxes allow false positives on irregular shapes (like the scissors), while strict thresholds punish valid actions on large elements (like the mouse). AgentVQA circumvents this by transforming the continuous coordinate space into a discrete decision problem, where distractors are programmatically ensured to be incorrect (e.g., outside the ground truth bounds) yet plausible enough to test precision.

# J   IMPLICATIONS AND SUGGESTIONS FOR PRACTITIONERS

AgentVQA provides unique diagnostic data unavailable in general VQA benchmarks. Based on the granular failure modes identified in our experiments, we offer the following concrete recommendations for future agentic model design and training:

- **Accelerate Model Iteration:** AgentVQA provides a scalable testbed for cross-domain performance. Evaluating 13,400 complex questions with a reasoning model judge (generating around 2,000 tokens/decision) would require 26.8 million output tokens for a single pass. AgentVQA mitigates these massive computational costs by enabling rapid, deterministic scoring for faster development cycles.
- **Bridge the Semantic Gap:** Our analysis reveals a "semantic-spatial disconnect" where models know what to do (semantic) but fail to locate where to do it (spatial)—for instance, models often correctly identify the next step in Web Agents but fail the precise coordinate grounding (46% error rate). This indicates that scaling model size is insufficient, and researchers must prioritize specialized grounding pre-training to link high-level intent with low-level execution.

- **Adopt Adaptive Inference:** Since reasoning models often underperform on perception tasks, we recommend adaptive architectures that decouple fast, robust perception modules (for spatial awareness) from expensive Chain-of-Thought (for planning) to prevent "Thinking Loops."

- **Implement Domain-Specific Curricula:** The lack of correlation between general VQA and agentic performance suggests generic tuning is inadequate. Our granular error profiles suggest a move toward domain-specific curriculum learning, targeting 3D spatial data for Robotics versus causal reasoning for Games.

- **Deploy Offline Safety Gates:** We propose using AgentVQA as a standardized diagnostic filter to detect critical reliability risks such as hallucinations and recursive loops before deploying agents into high-stakes, open-ended environments.

## K    ANALYSIS OF "THINKING LOOPS" AS A FAILURE MODE

During our analysis of different reasoning models, we identified some models like GLM-4.1V-Thinking and Kimi-VL-Thinking getting stuck in a repetitive cycle of reasoning without making any forward progress. For instance, when analyzing one model's reasoning process, we observed it getting stuck in the loop shown below:

```
<think> The person in the video is seen placing a small triangle on the table. The next shape
    that the person places is a large circle. The person then places another triangle next to
    the circle. The person then places another triangle next to the circle and the triangle.
    The person then places another triangle next to the circle and the triangle. The person
    then places another triangle next to the circle and the triangle. ...
```

## L    EVALUATION PROMPT

We use a standard, simple prompt structure that provides the model with the question, any relevant context such as action history, and concludes by instructing it to output only the single letter corresponding to its final choice. The sample prompt structure is mentioned below (Action History would be removed in case of non-trajectory based questions):

---

**Evaluation Prompt Template**

This is the standard template used for model evaluation. Placeholders like `{episode_goal}` are replaced with the specific data for each question.

---

**Goal:** `{episode_goal}`

**Action History:**
`{action_history if action_history.strip() else "No previous actions"}`

**Question:** `{question}`

**Options:**
`{options_text}`

Based on the provided screenshot, goal, action history, and question, select the single best option from the list above.
**IMPORTANT:** Your response must be EXACTLY one character (A, B, C, or D) with no other text, explanation, or formatting.

**Answer:**

---

## M  HARD NEGATIVE GENERATION PROMPT

This is the primary prompt template used to generate hard negative distractors for datasets like AITW; we use a similar but tailored schema for other datasets requiring transformation.

---

**Context:**

- The UI screenshot has dimensions `{width}x{height}` pixels.
- The image provided has a green dot marking the location of the correct action if it is a tap.
- The correct action is: **{correct_action_string}**

**Goal:** `{goal}`
**Previous Actions (History):**
`{history if history else "This is the first step."}`

---

**Your Task:**
Generate **three** distinct and plausible but **incorrect** distractor actions. These distractors should be designed to confuse a tester. Use a mix of the following strategies:

1. **Near-Miss:** An action of the same type as the correct one but slightly off (e.g., tapping right next to the correct button).
2. **Semantic-Confusion:** An action on a different but visually or functionally similar element (e.g., tapping 'Bluetooth Settings' when 'Wi-Fi Settings' is correct).

**Supported Action Formats (Use these formats EXACTLY):**

- `Tap: [x, y]` (where x and y are integer pixel coordinates)
- `Swipe: DIRECTION` (where DIRECTION is one of 'Up', 'Down', 'Left', 'Right')
- `Type: 'text to type'` (where 'text to type' is a plausible but incorrect string)
- `Button: ACTION` (where ACTION is one of 'Press Back', 'Press Home', 'Press Enter')

**Important Rules:**

- Do **NOT** generate the correct answer (`{correct_action_string}`).
- Ensure all tap coordinates are within the image bounds (width: `{width}`, height: `{height}`).
- For "Tap" distractors, analyze the image to choose locations that are genuinely incorrect but tempting.
- Generate a diverse set of three distractors.
- You also need to ensure that for near miss in tap the point is actually a negative.

Respond **ONLY** with a valid JSON object in the following format:

```
{"distractors": ["ACTION_TYPE: value", "ACTION_TYPE: value",
"ACTION_TYPE: value"]}
```

---

## N    ERROR MODE ANNOTATION PROMPT

The following prompt is used to have Gemini 2.5 Pro assign a specific error mode to a model's incorrect prediction; the prompt provides full context about the question and the model's failure, along with a detailed taxonomy of all possible error modes and their definitions.

---

You are an expert AI agent evaluator specializing in embodied robotics and robotic reasoning tasks. Your task is to analyze why a vision-language model failed on a specific question and categorize the failure.

---

### N.1    CONTEXT

QUESTION: {question}

QUESTION TYPE: {question_type}

NUMBER OF IMAGES: {num_images}

DATASET: {dataset_name}

### N.2    VISUAL ANALYSIS

Some description about the expected image scene.

### N.3    FAILURE DETAILS

MODEL'S PREDICTION: {model_prediction}
CORRECT ANSWER: {correct_answer}
MODEL'S REASONING: {reasoning (if available)}

### TASK

Analyze the model's failure in this embodied robotics reasoning task. Consider whether the model correctly understood the spatial relationships, identified objects and their attributes, predicted action outcomes, handled quantitative aspects, comprehended the instruction, and mapped its reasoning to the correct output. Focus on identifying the root cause using the specialized robotics error modes below.

### ERROR MODES

*{A detailed list of all error modes and their definitions is provided here.}*

---

Respond with a JSON object containing the 'error_mode' and a brief 'justification' for your choice. If you think that none of the error_modes apply then write NaN in error mode and justify in justification and also propose some new error mode in the justification and explain how this one aligns well with this sample as compared to others. It is not always necessary that an error mode applies so you can output NaN along with the justification and closest error mode from list and your idea of error mode for this.

