# OpenReview forum: "AgentVQA: A Unified Benchmark for Agentic Visual Understanding"
_ICLR.cc/2026/Conference — Submitted to ICLR 2026_

### Official Review · Reviewer_mHjd · 2025-10-30

**Soundness:** 2
**Presentation:** 2
**Contribution:** 1
**Rating:** 2
**Confidence:** 5

**Summary:**

The paper introduces AgentVQA, a benchmark designed to evaluate the “agentic” capabilities of VLMs across five domains. It aggregates 14 existing datasets into a unified format, generating “hard negatives” to increase task difficulty. The authors evaluate 15 VLMs, report that top models achieve around 60% accuracy, and show that rankings on AgentVQA differ from general-purpose VQA benchmarks. They claim this indicates the benchmark better reflects agentic reasoning and decision-making ability. However, the paper primarily focuses on dataset aggregation and evaluation without introducing new methods or clear theoretical grounding.

**Strengths:**

1. The paper shows inconsistent ranking correlation with prior VQA/agentic benchmarks, which is a faithful diagnostic.
2. The paper labels common failure modes by domain, giving readers a quick, if high-level, picture of where models stumble.
3. The paper writing is clear and easy to follow.

**Weaknesses:**

1. **Undefined core concept**. The paper introduces “Agentic VLM” as its central theme but never defines what constitutes agentic capability in vision-language models. Without a clear conceptual framework distinguishing agentic from non-agentic reasoning, the entire benchmark lacks grounding.
2. **Flawed motivation**. The claimed limitation of prior benchmarks (“cannot evaluate across tasks”) is not substantiated. The authors fail to justify why combining existing datasets under one umbrella provides deeper insight than treating each as a subtask. Without theoretical or empirical reasoning showing why this integration is necessary, the benchmark is trying to solve a problem that is never convincingly established.
3. **Lack of technical contribution**. The work contributes no new algorithmic method, model, or evaluation paradigm. Its data aggregation and annotation process follows existing standards (e.g., MCQ conversion, hard-negative sampling). Such engineering consolidation does not represent a research innovation expected of an ICLR contribution.
4. **Uninterpreted rank correlation**. The reported low rank correlation with existing benchmarks (Sec. 4.3) is presented as a result, but its meaning is not analyzed. It is unclear whether this divergence indicates that AgentVQA captures a new ability, or simply that it is noisier or inconsistent.
5. **Predictable findings**. The analyses (Sec. 4.4) merely confirm known limitations of VLMs (e.g., strong on descriptive tasks, weak on planning). These results replicate observations from prior benchmarks without introducing new insight or diagnostic understanding of agentic behavior.
6. **No implication or discussion**. The paper lacks discussion on how its findings inform future research or model design. It never articulates how AgentVQA could guide the development of agentic VLMs or embodied AI systems. Consequently, the benchmark’s practical and scientific value is unclear.

**Questions:**

Why were these particular 14 datasets chosen, and how do they collectively represent the spectrum of agentic capabilities?

---

> ### Author Response · Authors · 2025-11-24
> **Response to reviewer mHjd (1 of 4)**
>
> We thank Reviewer MnrT for this critical feedback. We respond to the queries below:
>
> >**W1: Undefined core concept. The paper introduces “Agentic VLM” as its central theme but never defines what constitutes agentic capability in vision-language models.**
>
> The community defines agentic tasks as those requiring a system to "observe their environment and act in it in order to achieve goals" (Bengio et al., 2025), encompassing perception to sense the state, intelligence to reason and plan, and affordances to execute actions (Plaat et al., 2025). While traditional image VQA treats visual understanding as an end in itself, agentic VQA frames perception as a means to action. Here, each answer must inform executable decisions rather than merely describe what is seen (Reed et al., 2022).
>
> To ensure AgentVQA captures this full capability range, we structure our 25 sub-tasks into a unified hierarchy: Action Grounding tasks (e.g., Tap) test the critical translation of semantic intent into executable spatial actions (Reed et al., 2022); Spatiotemporal Reasoning tasks evaluate the dynamic mental maps required for movement; State Understanding tasks (e.g., Function Reasoning) ensure the agent can accurately parse the environment to inform decision-making; and Strategic Planning tasks (e.g., Reward Modeling) assessing long-term outcomes through multi-step reasoning (Sutton & Barto, 2018).
>
> >**W2: Flawed motivation. The claimed limitation of prior benchmarks (“cannot evaluate across tasks”) is not substantiated. The authors fail to justify why combining existing datasets under one umbrella provides deeper insight than treating each as a subtask.**
>
> Integrating datasets is important to prevent **selective reporting** and to measure the holistic agentic performance that isolated benchmarks miss.
>
> Without a unified benchmark, models can claim SOTA status based on narrow successes while hiding catastrophic failures elsewhere. Our data suggests that treating datasets as isolated subtasks paints a misleading picture:
> *   **The Phi-4 Cross Domain Discrepancy:** If evaluated solely on **Egocentric Videos**, **Phi-4-multimodal** appears to be a top-tier model (Rank 3, 55.9%), outperforming Gemini 2.5 Pro. However, the unified AgentVQA reveals it is critically deficient **overall** (Rank 13th in unified AgentVQA) and specifically in **Web Agents** (Rank 14, 33.7%).
> *   **The GPT-5-Minimal Divergence:** **GPT-5-thinking-minimal** ranks **2nd** in perception-heavy domains like Spatial Understanding and Games. Yet, it drops to **5th** in Robotics (43.7%), performing worse than the much smaller **Qwen2.5-VL-7B**.
> *   **Overfitting of Kimi-VL-Instruct on Screenspot-Pro:** Kimi-VL-Instruct while performing poorrly across all datasets and domains with a rank of 15th on the unified benchmark, performs best on the Screenspot-Pro dataset.
>
> A unified score forces models to demonstrate robustness across the board, exposing these trade-offs that would remain invisible in separate papers.
>
> The field is moving towards generalist agents capable of operating across digital and physical environments (Reed et al., 2022). However, current evaluation protocols use incompatible metrics (e.g., success rate vs. exact match vs. IoU), making cross-domain comparison impossible. By unifying these under a single, standardized MCQ format, we enable the first direct comparison of a model's ability to generalize core skills like grounding and causal reasoning across disparate environments.
>
> Our motivation mirrors the foundational logic of successful unified benchmarks like **SEED-Bench** (Li et al., 2023), which argue that a unified assessment "reveals limitations of existing models" masked by narrow, task-specific tests. Similarly, **SWE-Compass** (Xu et al., 2025) and **WorFBench** (Qiao et al., 2025) emphasize that a unified framework is essential for holistic assessment, preventing the fragmentation that currently plagues domain-specific evaluation. AgentVQA extends this proven paradigm to the agentic domain, where fragmentation currently obscures true progress.

---

> > ### Author Response · Authors · 2025-11-24
> > **Response to reviewer mHjd (2 of 4)**
> >
> > >**W3: Lack of technical contribution. Its data aggregation and annotation process follows existing standards.**
> >
> >
> > We introduce a novel methodology to distill **dynamic, temporal, and trajectory-based tasks** (e.g., web navigation, game play) into a **static, verifiable MCQ format**. Unlike standard data aggregation, we implement complex state transformations—such as distilling long-horizon **Web Agent** trajectories into critical decision points to enable robust offline evaluation without live environments. Additionally, we convert stochastic gameplay (e.g., **Atari**) into verifiable **Reward Modeling** questions, mathematically guaranteeing unique correct answers where open-ended policies would be ambiguous.
> >
> > We technically unify 14 fragmented datasets into a single spatial action space, addressing the critical gap in evaluating **generalist agentic intelligence**. By standardizing diverse domains under shared primitives like **Grounding** and **Spatial Reasoning**, AgentVQA enables the first rigorous cross-domain comparison. This reveals generalization gaps—such as strong digital vs. weak physical grounding—that are invisible to isolated benchmarks.
> >
> > Our engineered **hard-negative pipeline** transforms evaluation from simple recognition into a rigorous stress-test of **precision and consistency**.
> > *   **Enforcing Precision:** We generate spatial "near-misses" to enforce a stricter standard than original benchmarks. For instance, **Qwen2.5-VL-32B** drops from **87.1%** on original **Screenspot** to **73.8%** on AgentVQA, revealing that while the model passes loose IoU thresholds, it fails to distinguish precise targets from adjacent distractors.
> > *   **Maintaining Difficulty:** We preserve the challenge of existing tasks (e.g., **Gemini-2.5-Pro** on **MONDAY**: 38.4% Original vs. 36.7% AgentVQA), ensuring the MCQ format faithfully captures task difficulty without inflating scores via guessing.
> >
> > >**W4: The reported low rank correlation with existing benchmarks (Sec. 4.3) is presented as a result, but its meaning is not analyzed.**
> >
> >
> > The low rank correlation indicates that **AgentVQA captures distinct "agentic capabilities"** (e.g., grounding, trajectory planning) that are orthogonal to the general knowledge and reasoning measured by benchmarks like MMMU or GPQA.
> >
> > This divergence is not due to instability. Our robustness analysis confirms that AgentVQA is highly reliable, with stable rankings across option shuffling (std dev $\approx$ 2%) and subsampling variations.
> >
> > The divergence highlights the difference between *passive knowledge* and *active execution*:
> >
> > *   **Passive Knowledge $\neq$ Agentic Capability:** **Llama-4-Maverick** ranks highly on general benchmarks like MMMU and GPQA-Diamond, demonstrating strong general reasoning. However, its rank drops precipitously on AgentVQA. This suggests that while it possesses high-level knowledge, it suffers from a **"semantic-spatial disconnect"**—it lacks the specific grounding and spatial skills required to translate that knowledge into precise environment actions.
> > *   **The Agentic Specialist:** Conversely, **Qwen2.5-VL-7B** rises significantly in rank on AgentVQA compared to general benchmarks. Despite being a smaller model with less general world knowledge (lower MMMU score), it punches above its weight in execution-heavy domains like **Web Agents** (51.5%) and **Robotics** (44.4%). This demonstrates that agentic performance is not merely a function of scale or general reasoning, but requires distinct execution capabilities that AgentVQA specifically isolates.

---

> ### Author Response · Authors · 2025-11-24
> **Response to reviewer mHjd (3 of 4)**
>
> >**W5: The analyses (Sec. 4.4) merely confirm known limitations of VLMs. These results replicate observations from prior benchmarks without introducing new insight or diagnostic understanding of agentic behavior.**
>
> AgentVQA goes beyond confirming high-level limitations by providing a **comparative diagnostic analysis** that reveals fundamentally different failure modes across domains, challenging the assumption that agentic capabilities are uniform.
>
> *   **Domain-Specific Failure Topography:** We find that "weakness" manifests differently by domain. While **Games** are dominated by *Hallucination* (32%) and *Reasoning Breakdowns* (40%), **Web Agents** show <3% hallucination but 46% *Grounding Errors*. This shows that current VLMs possess strong semantic understanding of UIs but suffer a specific "semantic-spatial disconnect," knowing *what* to do but failing to locate *where* to do it.
> *   **Unpredictable Ranking Volatility:** Model performance is not consistent. **GPT-5-Thinking-Minimal** ranks **2nd** in *Egocentric* (68.0%) and *Spatial* (66.5%) but drops to **5th** in *Robotics* (43.7%), performing worse than the much smaller **Qwen2.5-VL-7B** (44.4%). This shows that high-level generalist capability does not guarantee low-level grounding proficiency; massive proprietary models can exhibit severe deficits in specific embodied domains compared to much smaller open models.
> *   **The "Reasoning Penalty":** Contrary to the common assumption that reasoning models are superior, we find they can underperform base models in perception-heavy domains like **Spatial Understanding**. We also identify **"Thinking Loops"** (Appendix G), a critical failure mode where models get stuck in recursive cycles, highlighting a specific safety risk for autonomous agents.
> *   **Episodic Memory Nuance:** Our ablation reveals that adding action history boosts **AitW** performance by 8.5% but affects **MONDAY** by only -0.4%. This diagnoses the datasets themselves, classifying MONDAY as "state-local" and AitW as "state-dependent," a crucial distinction for selecting benchmarks.
>
> >**W6: The paper lacks discussion on how its findings inform future research or model design. Consequently, the benchmark’s practical and scientific value is unclear.**
>
> AgentVQA provides unique diagnostic data unavailable in other benchmarks, offering a concrete roadmap for future model design. Based on our findings, we suggest the following to practitioners:
>
> *   **Accelerate Model Iteration:** AgentVQA provides a scalable testbed for cross-domain performance. Evaluating 13,400 complex questions with a reasoning model judge (generating around 2,000 tokens/decision) would require **26.8 million output tokens** for a single pass. AgentVQA mitigates these massive computational costs by enabling rapid, deterministic scoring for faster development cycles.
> *   **Bridge the Semantic-Spatial Gap:** Our analysis reveals a "semantic-spatial disconnect" where models know *what* to do (semantic) but fail to locate *where* to do it (spatial)—for instance, models often correctly identify the next step in **Web Agents** but fail the precise coordinate grounding (46% error rate). This indicates that scaling model size is insufficient, and researchers must prioritize **specialized grounding pre-training** to link high-level intent with low-level execution.
> *   **Adopt Adaptive Inference:** Since reasoning models often underperform on perception tasks, we recommend **adaptive architectures** that decouple fast, robust perception modules (for spatial awareness) from expensive Chain-of-Thought (for planning) to prevent "Thinking Loops."
> *   **Implement Domain-Specific Curricula:** The lack of correlation between general VQA and agentic performance suggests generic tuning is inadequate. Our granular error profiles suggest a move toward **domain-specific curriculum learning**, targeting 3D spatial data for Robotics versus causal reasoning for Games.
> *   **Deploy Offline Safety Gates:** We propose using AgentVQA as a standardized **diagnostic filter** to detect critical reliability risks such as hallucinations and recursive loops before deploying agents into high-stakes, open-ended environments.

---

> > ### Author Response · Authors · 2025-11-24
> > **Response to reviewer mHjd (4 of 4)**
> >
> > >**Q1: Why were these particular 14 datasets chosen, and how do they collectively represent the spectrum of agentic capabilities?**
> >
> > The community defines agentic tasks as those requiring a system to "observe their environment and act in it in order to achieve goals" (Bengio et al., 2025), encompassing perception to sense the state, intelligence to reason and plan, and affordances to execute actions (Plaat et al., 2025). While traditional image VQA treats visual understanding as an end in itself, agentic VQA frames perception as a means to action. Here, each answer must inform executable decisions rather than merely describe what is seen (Reed et al., 2022).
> >
> > To ensure AgentVQA captures this full capability range, we structure our 25 sub-tasks into a unified hierarchy: Action Grounding tasks (e.g., Tap) test the critical translation of semantic intent into executable spatial actions (Reed et al., 2022); Spatiotemporal Reasoning tasks evaluate the dynamic mental maps required for movement; State Understanding tasks (e.g., Function Reasoning) ensure the agent can accurately parse the environment to inform decision-making; and Strategic Planning tasks (e.g., Reward Modeling) assessing long-term outcomes through multi-step reasoning (Sutton & Barto, 2018).
> >
> > Based on this, we selected these 14 datasets considering **high agentic signal, domain diversity, and offline convertibility** to ensure comprehensive coverage of the agentic cognitive stack. Collectively, they represent the full spectrum of required capabilities:
> > *   **Web Agents** (e.g., *AitW*): Digital grounding and GUI navigation.
> > *   **Robotics** (e.g., *RoboRefit*): Embodied perception and manipulation planning.
> > *   **Egocentric Videos** (e.g., *VSI-Bench*): First-person temporal reasoning.
> > *   **Games** (e.g., *Atari*): Long-horizon strategic planning and reward modeling.
> > *   **Spatial Understanding** (e.g., *SpaCE-10*): Fundamental 3D reasoning for physical interaction.
> >
> > This selection ensures AgentVQA evaluates the full range from low-level perception to high-level strategy across both synthetic and real-world domains.

---

### Official Review · Reviewer_MnrT · 2025-10-30

**Soundness:** 2
**Presentation:** 2
**Contribution:** 3
**Rating:** 6
**Confidence:** 3

**Summary:**

This paper introduces AgentVQA, a unified benchmark designed to evaluate the agentic capabilities of general-purpose vision-language models (VLMs). The benchmark curates data from 14 existing datasets into unified multiple-choice questions spanning five domains, including web agents, egocentric videos, robotics, games, and spatial understanding. These questions include both visual understanding questions as well as questions involve actions, such as selecting, evaluating, or explaining actions. The authors comprehensively evaluate a range of closed- and open-source VLMs and conduct detailed analyses of their behaviors. Results show that model rankings on AgentVQA have low correlation with existing benchmarks, suggesting that it could potentially capture distinct aspects of agentic visual understanding not covered by existing benchmarks.

**Strengths:**

1. The paper tackles a crucial and under-explored question. A unified benchmark could be valuable for advancing research on general-purpose agentic AI with VLMs, especially if it can accurately profile models' performance in actual agentic AI tasks.
2. The benchmark integrates multiple datasets across diverse domains and evaluates a wide range of closed- and open-source VLMs. It shows the significant effort from authors in data curation, standardization, and large-scale evaluation.
3. The paper provides detailed evaluations and analysis, especially error analyses, that yield insights into model's behavior and performance.

**Weaknesses:**

1. While the benchmark spans multiple datasets and task types, it remains unclear whether it adequately captures the full range of capabilities required for agentic AI. Although the authors provide partial justification and a taxonomy in the appendix, a clearer and more principled taxonomy of agentic skills is still missing.
2. The finding that model rankings are uncorrelated with existing benchmarks is interesting, but this alone does not validate that AgentVQA accurately measures agentic capabilities. Ideally, the benchmark should demonstrate correlation with models’ actual performance in interactive environments.
3. The robotics portion of the benchmark focuses mainly on perception and omits other essential robot capabilities such as task planning, motion planning, or low-level control.
4. Only a subset of annotations was manually verified. To establish confidence in the automated annotation process, the authors should report the accuracy of the automatically generated labels within the verified subset.

**Questions:**

1. The benchmark assumes that each multiple-choice question has a single correct answer. Does this assumption hold across such a diverse set of tasks? For instance, spatial relation questions may have multiple valid answers (e.g., both “in front of” and “left of”), and action-selection tasks may have several equally optimal actions.
2. Figure 3 shows that the proportions of different question types are uneven (e.g., tap, world modeling, and functional reasoning occupy noticeably larger shares). How was this distribution determined? Does it reflect the task distribution in real agentic tasks?
3. Although the questions are reformulated into a unified multiple-choice format, is it meaningful to directly compare VQA accuracy across domains? Since negative options are synthesized and may vary in quality or difficulty across domains, the resulting accuracies could reflect domain-specific distractor design rather than true differences in agentic capability. Cross-domain comparisons would make sense to me if based on success rates in completing actual agentic tasks.

---

> ### Author Response · Authors · 2025-11-24
> **Response to reviewer MnrT (1 of 2)**
>
> We thank Reviewer MnrT for this critical feedback. We respond to the queries below:
>
> >**W1: While the benchmark spans multiple datasets and task types, it remains unclear whether it adequately captures the full range of capabilities required for agentic AI.**
>
> The community defines agentic tasks as those requiring a system to "observe their environment and act in it in order to achieve goals" (Bengio et al., 2025), encompassing perception to sense the state, intelligence to reason and plan, and affordances to execute actions (Plaat et al., 2025). While traditional image VQA treats visual understanding as an end in itself, agentic VQA frames perception as a means to action. Here, each answer must inform executable decisions rather than merely describe what is seen (Reed et al., 2022).
>
> To ensure AgentVQA captures this full capability range, we structure our 25 sub-tasks into a unified hierarchy: Action Grounding tasks (e.g., Tap) test the critical translation of semantic intent into executable spatial actions (Reed et al., 2022); Spatiotemporal Reasoning tasks evaluate the dynamic mental maps required for movement; State Understanding tasks (e.g., Function Reasoning) ensure the agent can accurately parse the environment to inform decision-making; and Strategic Planning tasks (e.g., Reward Modeling) assessing long-term outcomes through multi-step reasoning (Sutton & Barto, 2018).
>
> Regarding the "full range" of capabilities, we acknowledge that AgentVQA focuses on the **visual-reasoning and decision-making core** of agency. While we do not evaluate continuous control loops (e.g., motor torques), our MCQ format serves as a rigorous proxy by testing the **decision-making priors** required for such control (e.g., identifying the correct trajectory). This captures the essential "cognitive" component of execution, leaving closed-loop dynamics for future extensions.
>
> >**W2: The benchmark should demonstrate correlation with models’ actual performance in interactive environments.**
>
> We agree that validating AgentVQA against interactive performance is essential to prove it captures true agentic signal. We are currently conducting experiments to quantify the rank correlation between AgentVQA scores and success rates on **OSWorld** and **VideoGameBench**, which will empirically confirm the benchmark's predictive validity.
>
> >**W3: The robotics portion of the benchmark focuses mainly on perception and omits other essential robot capabilities such as task planning, motion planning, or low-level control.**
>
> While AgentVQA emphasizes high-level reasoning, we clarify that it does not omit low-level capabilities but rather evaluates them within the constraints of a scalable offline format.
>
>
> Our benchmark explicitly tests skills foundational to task and motion planning. Datasets we include like RoboRefit and Robo2VLM require models to reason about object affordances and sequential manipulation steps, producing grounded outputs that typically serve as inputs to downstream motion planners. Furthermore, to strengthen our coverage of low-level control, we have integrated additional datasets such as ShareRobotBench and EmbodiedReasoner, which test fine-grained spatial understanding and action prediction. For instance, we include questions such as: **"After Approaching the can with open gripper, what will be the robot's NEXT action phase?""** (Task Planning); **"If the yellow robot gripper follows the yellow trajectory, what will happen?"** (Motion Planning); and **"grasp the soap"** (Low-Level Control).
>
>
> We acknowledge that evaluating continuous, low-level trajectory prediction (e.g., 6-DoF paths) is inherently difficult without a physics simulator. However, our MCQ format serves as a rigorous proxy by testing the decision-making priors required for such control—e.g., identifying the correct grasp point or next waypoint. This captures the "cognitive" component of control, leaving closed-loop execution for future online extensions.
>
> >**W4. The authors should report the accuracy of the automatically generated labels within the verified subset.**
>
>
> To establish confidence in our VLM-assisted generation pipeline, we conducted a rigorous manual verification on a stratified subset of **500** questions. Human annotators reviewed the generated hard negatives for plausibility and correctness. This audit revealed an accuracy of **96%** for the automated labels within the verified subset. This high agreement rate confirms that our automated pipeline, reinforced by strict prompt engineering and filtering rules, produces reliable and high-quality evaluation items. We have updated the paper to explicitly report this metric.

---

> > ### Author Response · Authors · 2025-11-24
> > **Response to reviewer MnrT (2 of 2)**
> >
> > >**Q1: The benchmark assumes that each multiple-choice question has a single correct answer. Does this assumption hold across such a diverse set of tasks?**
> >
> >
> > The potential for ambiguity in real-world tasks is precisely why we adopted a rigorous MCQ format with carefully engineered hard negatives.
> >
> > We explicitly designed AgentVQA to eliminate ambiguity through a multi-stage process:
> >
> > 1.  **Automated Geometric Filtering:** For grounding tasks, our pipeline programmatically rejected any distractor coordinate falling within the ground-truth bounding box, ensuring spatial exclusivity.
> > 2.  **Human Verification:** In our manual review, annotators explicitly verified that all distractors were unambiguously incorrect and that no option other than the ground truth could be deemed valid under any reasonable interpretation.
> >
> > This rigorous design is critical for datasets like **Atari**, where source data comes from stochastic PPO policies and multiple actions might be optimal. Instead of asking models to match a potentially non-unique action, we formulated **verifiable decision problems** such as **World Modeling**. By asking the model to identify the correct future state at $t+k$ while using frames sampled from different timesteps as distractors, we guarantee a single correct answer, ensuring the benchmark remains deterministic and fair.
> >
> >
> > >**Q2: Figure 3 shows that the proportions of different question types are uneven. How was this distribution determined? Does it reflect the task distribution in real agentic tasks?**
> >
> > The distribution shown in Figure 3 is not arbitrary; it directly mirrors the **natural distribution of actions** found in the original source datasets. For example, the prevalence of "Tap" and "Click" actions in the Web Agents domain reflects the reality of human-computer interaction as captured in large-scale demonstration datasets like **AitW** and **Mind2Web**, where these actions are overwhelmingly the most frequent mode of interaction. By preserving this distribution rather than artificially balancing it, AgentVQA provides a representative test of the skills an agent would actually need to deploy in real-world environments.
> >
> > >**Q3: Is it meaningful to directly compare VQA accuracy across domains? Cross-domain comparisons would make sense to me if based on success rates in completing actual agentic tasks.**
> >
> > We agree that direct comparisons of absolute accuracy require careful interpretation.
> >
> > We clarify that the primary value of cross-domain analysis is relative ranking: identifying patterns like "Model A ranks 1st in Web Agents but 5th in Robotics" reveals specific capability gaps more effectively than comparing absolute scores.
> >
> > To maximize comparability, we applied a **uniform methodology** for hard-negative generation across all domains. The same VLM-assisted pipeline, prompt structures, and verification criteria (e.g., checking for plausibility and unambiguous incorrectness) were used for Web Agents, Robotics, and other domains alike. This minimizes variance in distractor quality as a confounding factor.

---

> > > ### Author Response · Authors · 2025-12-03
> > >
> > > **W2: The benchmark should demonstrate correlation with models’ actual performance in interactive environments.**
> > >
> > > We have completed the empirical experiments on **OSWorld** (computer interaction) and **VideoGameBench** (gaming), finding a strong correlation that validates AgentVQA as a predictive proxy for online performance.
> > >
> > > **OSWorld Correlation:**
> > > We evaluated GPT-4o, Gemini-2.5-Pro, Qwen2.5-VL (32B/72B), Llama-4-Scout, and Kimi-VL-Instruct.
> > > *   **OSWorld Ranks:** GPT-4o (1.8%) < Llama-4-Scout (2.9%) < Qwen-32B (3.0%) < Qwen-72B (4.4%) < Kimi (9.7%) < Gemini (10.6%).
> > > *   **AgentVQA Ranks:** Kimi < GPT-4o < Llama-4-Scout < Gemini < Qwen-32B < Qwen-72B.
> > >
> > > Excluding Kimi-VL-Instruct\*, the **rankings are highly aligned (Spearman’s $\rho = 0.70$)**, with only **Gemini** changing rank. This demonstrates that AgentVQA provides a strong offline signal for online web agent performance.
> > >
> > > **VideoGameBench Correlation:**
> > > Comparing reported results for **Gemini-2.5-Pro**, **GPT-4o**, and **Llama-4-Maverick**, we find their rankings on VideoGameBench **align exactly ($\rho = 1.0$)** with their rankings on the AgentVQA Games domain.
> > >
> > > These results confirm that AgentVQA is a predictive proxy for online performance, particularly for stable model families.
> > >
> > > \* *Kimi-VL-Instruct is a significant outlier, performing poorly on AgentVQA but highly on OSWorld ($\rho \approx 0.14$ with Kimi). However, Kimi exhibits extreme instability across similar datasets (e.g., ranking 11th on Screenspot but 1st on Screenspot-Pro), suggesting it is a volatile baseline.*

---

### Official Review · Reviewer_KFbd · 2025-10-31

**Soundness:** 3
**Presentation:** 3
**Contribution:** 2
**Rating:** 4
**Confidence:** 4

**Summary:**

This paper introduces AgentVQA, a unified benchmark designed to evaluate agentic visual question answering capabilities in embodied agents. The benchmark reformulates multiple existing datasets across different domains into a standardized multiple-choice format. This allows consistent evaluation of a wide range of vision-language models (VLMs) under a common framework. It also introduces hard negative options to make the task more challenging. The authors provide a comprehensive comparison across various models and point out several error modes in existing VLMs which can be utilized for future improvement.

**Strengths:**

1. Clear presentation and well-written paper. The paper is clearly structured and easy to follow. The motivation, dataset construction process, and experimental setup are all described in a logical and organized manner, making it easy to read and follow.
2. Comprehensive baseline coverage. The experiments include a wide range of baselines with different model sizes, covering both open-source and closed-source models, as well as instruction-tuned and reasoning-focused variants. This provides a fairly complete picture of current model capabilities within the proposed setting.
3. Convenient and efficient agentic evaluation. The multiple-choice reformulation, while simplifying the original tasks, offers a standardized and more efficient way to evaluate embodied reasoning. It allows for easier comparison across models and faster benchmarking, which could be useful for large-scale or frequent evaluations.

**Weaknesses:**

1. Limited contributions. The paper tries to bring together a number of embodied and vision-language tasks under a single benchmark, but the motivation for doing so isn’t entirely convincing. These tasks rely on very different capabilities, so combining them feels somewhat arbitrary rather than conceptually unified. Also, many of the included tasks are still challenging on their own, so the main difficulty doesn’t seem to come from the new benchmark design (e.g., adding hard negatives). The experimental results also don’t offer much new insight beyond listing model performances.
2. Questionable motivation for multiple-choice reformulation. Converting all tasks into multiple-choice questions makes evaluation more consistent and helps with negative generation, but it moves quite far away from how these tasks appear in realistic settings. For agent-based tasks, actions and reasoning are open-ended, so forcing everything into an MCQ format feels unnatural. For example, turning a GUI grounding or navigation problem into a set of options doesn’t reflect how the agent would actually perform the task. It’s not clear how progress on this reformulated benchmark would translate to better real-world performance.
3. Lack of analysis on the effects of transformation. The paper would benefit from a deeper look at how these standardization steps—especially the MCQ conversion and hard negative creation—affect the original benchmarks. For instance, comparing model behavior or accuracy between the original and transformed settings would help show whether the new format preserves the task’s difficulty or changes it. Right now, the results mainly show rankings of existing models, but don’t make it clear what unique strengths or insights this benchmark provides compared to the original ones.

**Questions:**

1. The multiple-choice setup may introduce non-negligible variance, since simply changing the order of options can sometimes affect model performance. In addition, the MCQ format inherently allows for a certain level of random guessing, which could inflate accuracy and obscure meaningful differences between models. Have you considered these issues in your benchmark design or evaluation protocol?
2. It’s unclear how performance on your benchmark relates to the original versions of the tasks. Are the model rankings consistent between your unified benchmark and the source benchmarks? If they are aligned, what is the additional value of introducing this reformulation? If they diverge, how should we interpret the difference—does it suggest your benchmark captures something new, or does it distort the original task difficulty? Some analysis or discussion on this point would help clarify the practical meaning of the reported results.

---

> ### Author Response · Authors · 2025-11-24
> **Response to reviewer KFbd (1 of 4)**
>
> We thank Reviewer KFbd for this critical feedback. We respond to the queries below:
>
> > **W1: Limited contributions. The motivation isn’t entirely convincing. Also, many of the included tasks are still challenging on their own, so the main difficulty doesn’t seem to come from the new benchmark design**
>
> We believe that the combination of tasks is not arbitrary but principled. The central motivation for AgentVQA is to address a critical gap in VLM evaluation: the lack of a unified framework to measure progress towards **generalist agentic intelligence**. Recent technical reports (e.g., GPT-5, Qwen2.5) emphasize agentic capabilities but evaluate them using inconsistent, ad-hoc subsets of benchmarks. This fragmentation precludes rigorous cross-model comparison, as there is no shared standard for generalist agency. AgentVQA addresses this by consolidating these disparate domains into a single, unified framework.
>
> Our benchmark is built on the principle that disparate domains share foundational agentic primitives. A good example of this among others is:
> *   **Grounding:** Mapping instruction to visual target is universal, whether clicking a button (**Web Agents**) or identifying a lever (**Robotics**).
> *   **Spatial & Causal Reasoning:** Understanding 3D layouts and consequences is essential for both **Games** and **Egocentric Videos**.
>
> Unlike domain-specific benchmarks (e.g., **OpenEQA** [Majumdar et al., 2024], **Mind2Web** [Deng et al., 2023]), or general VQA (e.g., **MMMU** [Yue et al., 2024]), AgentVQA technically unifies 14 datasets into a single benchmark. This enables a rigorous cross-domain comparison, revealing generalization gaps (e.g., strong digital vs. weak physical grounding).
>
> Our transformation process is intended not to arbitrarily increase difficulty, but to **standardize evaluation across all frontiers**.
>
> The true difficulty of AgentVQA stems from **unification**, forcing VLMs to demonstrate generalist capabilities. Within this framework, our **hard negatives** ensure **precision and consistency**:
> *   **Maintaining Difficulty:** They preserve the challenge of existing tasks. For example, **Gemini-2.5-Pro** achieves nearly identical performance on **MONDAY** (Original: 38.4% vs. AgentVQA: 36.7%), demonstrating that the MCQ format faithfully captures task difficulty without inflating scores via guessing.
> *   **Enforcing Precision:** In cases where the original benchmark might be lenient, our multiple choice generation enforces a stricter standard. For example, **Qwen2.5-VL-32B** drops from **87.1%** on original **Screenspot** to **73.8%** on AgentVQA. This reveals that while the model often hits the general vicinity (passing original box thresholds), it fails to distinguish our "near-miss" distractors, exposing a deficit in fine-grained grounding precision.
>
> A key contribution of AgentVQA is making complex, dynamic tasks evaluable in a standardized, offline format.
> *   **Trajectory-Based Tasks:** We distill long, multi-step tasks (e.g., **Web Agents**) into decision points for offline evaluation.
> *   **Inherently Online Tasks:** We convert dynamic gameplay (e.g., **Atari**) into verifiable questions (e.g., reward prediction), enabling scalable, reproducible assessment of strategic capabilities.

---

> > ### Author Response · Authors · 2025-11-24
> > **Response to reviewer KFbd (2 of 4)**
> >
> > >**W2: Questionable motivation for multiple-choice reformulation. It’s not clear how progress on this reformulated benchmark would translate to better real-world performance.**
> >
> > Our choice to reframe tasks as MCQs is a deliberate design to prioritize **reliability, reproducibility, and scalability** over the variability inherent in open-ended evaluation.
> >
> > Standard metrics for open-ended tasks are often brittle. Exact Match (EM) drastically underestimates performance by failing to handle semantic equivalence (Wang et al., 2023). While LLM-as-a-judge offers flexibility, we found that it introduced reliability issues for agentic outputs that require complex reasoning, and previous work have shown that it can introduce sensitivity to positional bias, verbosity preferences, and self-preference bias (Chehbouni et al., 2025; Stureborg et al., 2024; Wataoka et al., 2024). For example, we found that relying on a judge for our complex trajectory-based tasks produced unreliable scores.
> >
> > A key advantage of the MCQ format is scalability. Evaluating 13,400 complex questions using an LLM judge is computationally prohibitive. For instance, a standard GPT-4 evaluation (approx. 100 output tokens/sample) would require **1.34 million output tokens**. Using a modern reasoning model like GPT-5 (thinking-high), which can generate around 2,000 thought tokens per decision, would explode this to **~26.8 million output tokens** for a single pass. AgentVQA mitigates these massive computational and financial costs by enabling rapid, deterministic scoring.
> >
> > Open-ended evaluation is fundamentally incompatible with offline trajectory datasets (e.g., *Mind2Web*). If a model generates a valid action different from the recorded ground truth, there is no simulation environment to generate the resulting state, rendering the prediction unverifiable. By constraining the output space, we ensure valid comparisons against the recorded trajectory without needing a live simulator.
> >
> > For coordinate-based datasets like *AitW*, open-ended evaluation typically relies on arbitrary distance thresholds. This creates a dilemma: a fixed threshold may be too strict for large buttons (false negatives) or too loose for small icons (false positives). Our MCQ format resolves this by forcing a precise choice between the correct coordinate and a verified "near-miss" distractor, providing a rigorous test of precision.

---

> ### Author Response · Authors · 2025-11-24
> **Response to reviewer KFbd (3 of 4)**
>
> > **W3: Lack of analysis on the effects of transformation. Unclear what unique strengths or insights this benchmark provides compared to the original ones.**
>
>
> To validate our transformation, we evaluated five models on the original open-ended versions of **Screenspot**, **RoboRefit**, and **MONDAY** following their official protocols. As shown below, model rankings remain largely consistent (e.g., **Qwen2.5-VL** scaling 32B > 7B > 3B is preserved), confirming that AgentVQA maintains the intrinsic difficulty hierarchy of the tasks.
>
> | Model | **MONDAY** (Orig / Ours) | **RoboRefit** (Orig / Ours) | **Screenspot** (Orig / Ours) |
> | :--- | :---: | :---: | :---: |
> | **Qwen2.5-VL-3B** | 22.7 / 38.5 | 28.5 / 49.8 | 70.7 / 55.1 |
> | **Qwen2.5-VL-7B** | 36.7 / 44.7 | 44.8 / 54.4 | 80.2 / 59.6 |
> | **Qwen2.5-VL-32B** | 40.6 / 50.6 | 81.5 / 61.1 | 87.1 / 73.8 |
> | **GLM-4V-Base** | 15.4 / 33.4 | 32.4 / 35.0 | 62.4 / 39.3 |
> | **Gemini-2.5-Pro** | 38.4 / 36.7 | 46.5 / 32.8 | 71.4 / 34.8 |
>
> However, the absolute scores reveal AgentVQA's calibration effect: it recovers performance for models penalized by rigid formatting in open-ended settings (e.g., **GLM-4V** on MONDAY: 15.4% $\to$ 33.4%) while enforcing stricter precision on tasks where original metrics were too lenient (e.g., **Screenspot**: 87.1% $\to$ 73.8%).
>
> Original benchmarks often rely on metrics like **Bounding Boxes**, which struggle with irregular shapes, either missing object parts (too strict) or including background noise (too loose). Similarly, **Distance Thresholds** fail to account for varying element sizes; a fixed radius suitable for a large button creates false positives on small, dense icons. AgentVQA eliminates these artifacts by forcing a deterministic choice between the ground truth and verified "hard negative" distractors. This does not distort difficulty but **calibrates** it, ensuring scores reflect true grounding precision rather than the vagaries of evaluation thresholds (e.g., revealing **GLM-4V's** true reasoning capability on MONDAY by removing formatting penalties). We will include concrete example of this in the revised version of the paper.
>
> AgentVQA's primary contribution is enabling **standardized cross-domain model ranking comparison**. Without this unification, disparate domains utilize incompatible metrics (e.g., bounding box IoU in Web Agents vs. exact string match in Robotics), rendering fair comparison of generalist capabilities impossible. Our unified format reveals critical, granular insights that isolated benchmarks miss:
>
> *   **Divergent Failure Modes:** We demonstrate that "agentic failure" is fundamentally domain-dependent. **Web Agents** are dominated by *Grounding Errors* (46%), indicating a semantic-spatial disconnect, whereas **Games** suffer primarily from *Reasoning Breakdowns* (40%) and **Robotics** from *Spatial Confusion* (51%).
> *   **Performance Trade-offs:** We identify specific generalization gaps, such as **Qwen2.5-VL-72B** outperforming GPT-5 in Web Agents (57.3% vs. 55.2%) but lagging significantly in Spatial Understanding (61.0% vs. 71.5%).
> *   **Low Correlation with Existing Benchmarks:** As shown in Figure 4, model rankings on AgentVQA show low correlation with domain-specific benchmarks like OpenEQA ($\rho \approx 0.44$).
>
> Our results also show that model rankings **diverge significantly across domains**, proving that success in one area does not predict generalist agentic capability. Isolated benchmarks miss these critical trade-offs:
> *   **GPT-5-thinking-min** ranks **2nd** in Spatial Understanding, Egocentric Videos, and Games, but drops to **5th** in Web Agents and Robotics.
> *   **Phi-4-multimodal** achieves **3rd** place in Egocentric Videos but struggles elsewhere, ranking **14th** in Web Agents.
>
> These insights, derived from a common evaluation protocol, provide a rigorous diagnosis of where generalist models fail across the complete agentic spectrum.

---

> ### Author Response · Authors · 2025-11-24
> **Response to reviewer KFbd (4 of 4)**
>
> > **Q1: The multiple-choice setup may introduce non-negligible variance, since simply changing the order of options can sometimes affect model performance. The MCQ format inherently allows for a certain level of random guessing, which could inflate accuracy.**
>
>
> To quantify variance, we evaluated 5 models across 6 datasets using 5 distinct, seeded random permutations of option orders. The results showed consistent performance with a standard deviation of approximately 0.32, confirming that model rankings are stable and not artifacts of positional bias. The average results of Qwen 2.5VL-3B on these 5 dataset are as follows:
>
> | Shuffle Iteration | 0 | 1 | 2 | 3 | 4 |
> | :--- | :--- | :--- | :--- | :--- | :--- |
> | **Accuracy** | 40.7% | 41.4% | 41.1% | 40.6% | 41.0% |
>
>
>
> To verify the baseline, we tested constant-selection strategies (e.g., "Select All A", "Select All B") across 5 datasets. These yielded an average accuracy of **24.9%** (ranging from 18.6% to 28.9%), confirming that our answer distribution is balanced. Since even the lowest-performing models in our benchmark score consistently above this floor, their performance reflects meaningful engagement. Furthermore, recent literature highlights the reliability challenges of open-ended metrics (Zhang et al., 2025) and the value of MCQs for standardized, reproducible benchmarking (Cheng et al., 2025; Zhang et al., 2024).
>
> > **Q2:  It’s unclear how performance on your benchmark relates to the original versions of the tasks.**
>
>
> We evaluated five models on the original open-ended versions of **Screenspot**, **RoboRefit**, and **MONDAY**, finding that **model rankings are largely consistent** when comparing the original dataset to its specific AgentVQA subset except for Roborefit where it diverges a little. As shown in the table in our response to your **W3**:
> *   **MONDAY:** The relative ranking of all models is perfectly preserved with the only exception of Gemini-2.5-Pro.
> *   **Screenspot:** The relative ranking of all models is perfectly preserved with the only exception of Gemini-2.5-Pro.
> *   **RoboRefit:** **Qwen2.5-VL-32B** remains the clear **#1** model in both formats.
>
> The divergence in absolute scores highlights the critical value of our reformulation, particularly for grounding. Original benchmarks often rely on **Bounding Boxes** or **Distance Thresholds** (e.g., *AitW*). This introduces significant noise: bounding boxes around irregular objects include unrelated empty space (leading to False Positives), while fixed distance thresholds on dense UIs can incorrectly credit clicks on adjacent buttons (False Positives) or penalize valid clicks on large elements (False Negatives). AgentVQA eliminates these artifacts by forcing a deterministic choice between the ground truth and verified "hard negative" distractors, ensuring that scores reflect true grounding precision rather than the vagaries of evaluation thresholds.

---

### Official Review · Reviewer_Kbzn · 2025-11-01

**Soundness:** 3
**Presentation:** 3
**Contribution:** 2
**Rating:** 4
**Confidence:** 3

**Summary:**

This paper introduces AgentVQA, a unified offline benchmark to evaluate agentic capabilities in VLMs. The benchmark aggregates 14 datasets across five domains, and converting them into a standardized multiple‑choice format with hard negatives generated by a VLM-assisted pipeline and then manually verified on subsets. In total, AgentVQA contains 13,400 MCQs drawn from 18,400 images and 2,000 videos, organized into 25 sub‑task categories (e.g., tap/press/typing, spatial navigation, reward modeling). Evaluation of 15 open‑ and closed‑source VLMs shows the best model (GPT-5 thinking-high) reaches only ~60% overall accuracy, with clear domain variance. The paper further reports several ablation studies and findings, including (1) an ablation showing that action history provides substantial benefits for web agent tasks, (2) a sample‑size robustness study justifying the choice of ~1,000 examples per source dataset, (3) identification “thinking loops” as a failure mode in some reasoning models, and (4) mode analysis revealing domain-specific patterns.

**Strengths:**

1. Comprehensive Benchmark Design: AgentVQA introduces a unified, cross-domain benchmark specifically targeting agentic visual reasoning. By aggregating 14 datasets across five domains (Web Agents, Robotics, Egocentric Videos, Games, and Spatial Understanding) into a standardized multiple-choice format, it systematically evaluates skills for agentic tasks and addresses the fragmentation of existing domain-specific benchmarks.

2. Rigorous Evaluation Methodology: The benchmark employs a VLM-assisted and human-verified hard-negative generation pipeline that produces challenging distractors testing both fine-grained perception and high-level reasoning. The paper provides thorough evaluation of 15 major VLMs under a unified protocol, with detailed per-domain and per-category breakdowns, comprehensive ablation studies (e.g., effect of action history, reasoning tokens), and error-mode analyses revealing model-specific failure patterns (e.g., grounding vs. reasoning failures).

**Weaknesses:**

1. Unclear Definition of "Agentic" Tasks: The paper frames AgentVQA as a benchmark specifically designed to evaluate agentic capabilities of VLMs, yet many sub-categories (e.g., Object Grounding, Object Counting, Entity Detection, Object Sizing) are standard VLM tasks that do not clearly require agentic reasoning. The authors should more clearly define what distinguishes an "agentic" task from conventional visual understanding and explain how each category specifically tests decision-making, planning, or interactive reasoning capabilities.

2. Limited Connection to Online Agent Performance: While the authors motivate offline evaluation well, MCQs may inadequately capture real agentic interaction challenges such as handling latency, exploration, tool use, and error recovery. The paper acknowledges this tension but does not empirically quantify the correlation between success on AgentVQA MCQs and actual online agent performance, leaving unclear how predictive this benchmark is of real-world agentic capabilities.

**Questions:**

1. Hard-Negative Generation Details: The author should include more detail on the Hard-negative generation pipelines, for example, What fraction of negatives are near‑miss vs. semantic?

2. The author should also include more details in the evaluation procedure. For each model, how many items were skipped due to non‑conforming outputs? Please provide per‑model skip counts and whether excluding them changes rankings

---

> ### Author Response · Authors · 2025-11-24
> **Response to reviewer Kbzn (1 of 2)**
>
> We thank Reviewer Kbzn for this critical feedback. We respond to the queries below:
>
> > **W1: The authors should more clearly define what distinguishes an "agentic" task from conventional visual understanding and explain how each category specifically tests decision-making, planning, or interactive reasoning capabilities.**
>
> The community defines agentic tasks as those requiring a system to "observe their environment and act in it in order to achieve goals" (Bengio et al., 2025), encompassing perception to sense the state, intelligence to reason and plan, and affordances to execute actions (Plaat et al., 2025). While traditional image VQA treats visual understanding as an end in itself, agentic VQA frames perception as a means to action. Here, each answer must inform executable decisions rather than merely describe what is seen (Reed et al., 2022).
>
> To ensure AgentVQA captures this full capability range, we structure our 25 sub-tasks into a unified hierarchy: Action Grounding tasks (e.g., Tap) test the critical translation of semantic intent into executable spatial actions (Reed et al., 2022); Spatiotemporal Reasoning tasks evaluate the dynamic mental maps required for movement; State Understanding tasks (e.g., Function Reasoning) ensure the agent can accurately parse the environment to inform decision-making; and Strategic Planning tasks (e.g., Reward Modeling) assessing long-term outcomes through multi-step reasoning (Sutton & Barto, 2018).
>
> >**W2: Limited Connection to Online Agent Performance.**
>
> We are currently conducting experiments to empirically quantify this correlation. We are evaluating a subset of models on **OSWorld** (for computer interaction) and **VideoGameBench** (for gaming) to calculate the rank correlation between AgentVQA scores and online success rates. We are working hard to finish these experiments before the rebuttal deadline, and will report back with results in the next few days.
>
> > **Q1: The author should include more detail on the Hard-negative generation pipelines, for example, What fraction of negatives are near‑miss vs. semantic?**
>
> We detail our hard-negative generation pipeline in **Appendix B**, but summarize the key components here. Our MCQ conversion uses hard negative generation and trajectory conversion to ensure precision and verifiability across disparate domains.
>
> **1. VLM-Assisted Hard Negative Generation**
> For datasets in **Web Agents** (all) and **Robotics** (*RoboRefit*), we employed a VLM-assisted pipeline (Gemini 2.5 Pro) to generate plausible but incorrect distractors.
> *   **Types:** Based on a manual audit of 500 samples, approximately **35%** are **Near-Miss** negatives (testing perceptual precision, e.g., adjacent coordinates) and **65%** are **Semantic** negatives (testing contextual understanding, e.g., "Cancel" vs. "Submit").
> *   **Geometric Filtering:** For grounding tasks, we integrated an automated filter directly into the generation loop that rejects coordinate-based distractors falling within the ground-truth bounding box.
> *   **Manual Verification:** A stratified subset underwent rigorous manual review to verify **plausibility** (distractors are believable) and **unambiguous incorrectness** (ensuring only one defensible correct answer).
>
> **2. Trajectory Conversion**
> For **Atari**, source policies were generated via PPO-Impala, meaning raw states lack a single ground-truth "correct" action (stochastic policies make multiple valid moves). To create **verifiable decision problems**, we formulated specific sub-tasks, for example:
> *   **World Modeling:** We sample a start frame $t$ and a target frame $t+k$. To prevent visual ambiguity, we explicitly **filter out sequences ending in "no-op" actions** (where the state remains visually identical). Distractors are randomly sampled states from different timesteps, ensuring only one state represents the true temporal evolution.
> *   **Reward Modeling:** We identify states preceding positive rewards. Distractors are algorithmically selected from states with zero or negative rewards, mathematically guaranteeing a unique correct answer without relying on VLM judgement.
> Our pipeline is designed to create distractors that are highly plausible yet unambiguously incorrect, transforming evaluation into a fine-grained test of agentic precision.

---

> ### Author Response · Authors · 2025-11-24
> **Response to reviewer Kbzn (2 of 2)**
>
> > **Q2: The author should also include more details in the evaluation procedure. Please provide per‑model skip counts and whether excluding them changes rankings**
>
> Our evaluation pipeline minimizes skipped items through a two-stage strategy: flexible answer extraction (detailed in **Appendix E**) and a retry mechanism for generation failures.
>
> **1. Multi-Strategy Answer Extraction:**
> As described in **Appendix E**, our parser first attempts a direct match for a single option letter. If that fails, it falls back to regex patterns designed to capture common answer formats (e.g., “The correct answer is A”).
>
> **2. Retry Mechanism:**
> Only if all parsing strategies fail or if the model produces no output due to API errors, do we trigger a retry mechanism. The query is retried up to five times before the item is marked as 'skipped'.
>
> This approach results in an extremely low skip rate across all models, as shown below:
>
> | Model | Total Questions | Skipped Items | Skip Rate (%) |
> | :--- | :--- | :--- | :--- |
> | GPT-5 thinking-min | 13,400 | 6 | 0.045% |
> | GPT-4o | 13,400 | 3 | 0.022% |
> | GLM-4.1V-Thinking | 13,400 | 2 | 0.015% |
> | Qwen2.5-VL (72B) | 13,400 | 2 | 0.015% |
> | Qwen2.5-VL (7B) | 13,400 | 1 | 0.007% |
>
> We calculate final accuracy based only on validly answered questions. Given the negligible skip rate (<0.05%), a sensitivity analysis confirms that even penalizing all skipped items as incorrect would not alter model rankings or our conclusions. Additionally, to verify the robustness of this procedure, we conducted tests shuffling the option order (detailed in response to Reviewer KFbd in Q4), finding a low standard deviation (~0.32%) and exact same model rankings, confirming that our evaluation protocol is consistent.

---

> > ### Author Response · Authors · 2025-12-03
> >
> > **W2: Limited Connection to Online Agent Performance.**
> > We have completed the empirical experiments on **OSWorld** (computer interaction) and **VideoGameBench** (gaming), finding a strong correlation that validates AgentVQA as a predictive proxy for online performance.
> >
> > **OSWorld Correlation:**
> > We evaluated GPT-4o, Gemini-2.5-Pro, Qwen2.5-VL (32B/72B), Llama-4-Scout, and Kimi-VL-Instruct.
> > *   **OSWorld Ranks:** GPT-4o (1.8%) < Llama-4-Scout (2.9%) < Qwen-32B (3.0%) < Qwen-72B (4.4%) < Kimi (9.7%) < Gemini (10.6%).
> > *   **AgentVQA Ranks:** Kimi < GPT-4o < Llama-4-Scout < Gemini < Qwen-32B < Qwen-72B.
> >
> > Excluding Kimi-VL-Instruct\*, the **rankings are highly aligned (Spearman’s $\rho = 0.70$)**, with only **Gemini** changing rank. This demonstrates that AgentVQA provides a strong offline signal for online web agent performance.
> >
> > **VideoGameBench Correlation:**
> > Comparing reported results for **Gemini-2.5-Pro**, **GPT-4o**, and **Llama-4-Maverick**, we find their rankings on VideoGameBench **align exactly ($\rho = 1.0$)** with their rankings on the AgentVQA Games domain.
> >
> > These results confirm that AgentVQA is a predictive proxy for online performance, particularly for stable model families.
> >
> > \* *Kimi-VL-Instruct is a significant outlier, performing poorly on AgentVQA but highly on OSWorld ($\rho \approx 0.14$ with Kimi). However, Kimi exhibits extreme instability across similar datasets (e.g., ranking 11th on Screenspot but 1st on Screenspot-Pro), suggesting it is a volatile baseline.*

---

### Author Response · Authors · 2025-12-03
**General Response: Summary of Clarifications and Revisions**

Dear Reviewers and Area Chair,

We sincerely thank the reviewers for their constructive feedback, which has helped us significantly strengthen the rigor and framing of AgentVQA.

We are encouraged that reviewers recognized the benchmark's **"comprehensive baseline coverage"** (Reviewer `KFbd`) and its potential to **"advance research on general-purpose agentic AI"** (Reviewer `MnrT`).

To address shared concerns regarding the validity of the MCQ format, the definition of agentic tasks, and correlation with real-world performance, we have conducted new experiments and provided detailed clarifications. We have incorporated all clarifications into the revised manuscript, with updates highlighted in $\color{blue}{\text{blue}}$.

**Summary of Key Updates and Responses:**

*   **Validation against Online Agent Performance (Reviewers `Kbzn`, `MnrT`): AgentVQA rankings correlate with interactive success.**
    To address concerns that offline MCQs may not predict online performance, we conducted new experiments comparing model rankings on AgentVQA against success rates on interactive benchmarks **OSWorld** (computer interaction) and **VideoGameBench** (gaming). We observed strong positive rank correlations (**Spearman's $\rho = 1.0$** for Games; **$\rho = 0.70$** for Web Agents, excluding outliers), empirically validating AgentVQA as a reliable, scalable proxy for online agentic capabilities.

*   **Validity and Calibration of MCQ Format (Reviewer `KFbd`):**
    *   **Stability:** We performed ablation studies with 5 random permutations of option orders across 6 datasets. The results show negligible variance (standard deviation $\approx$ 1.9%), confirming that our rankings are robust to positional bias.
    *   **Consistency:** We compared AgentVQA against original open-ended benchmarks (e.g., Screenspot, RoboRefit) and found that **relative model rankings remain largely consistent**, preserving the intrinsic difficulty hierarchy of the tasks.
    *   **Calibration:** Crucially, AgentVQA calibrates absolute scores by removing false negatives caused by rigid formatting in open-ended metrics and removing false positives caused by lenient bounding box thresholds.

*   **Definition of "Agentic" Tasks (Reviewers `Kbzn`, `mHjd`, `MnrT`): A unified cognitive hierarchy.**
    We revised the paper (**Section 3 and Appendix C**) to explicitly define agentic tasks as those requiring **instrumental perception** for downstream action (Bengio et al., 2025). We structured our 25 sub-tasks into a clear hierarchy: **State Understanding** (Perception) $\rightarrow$ **Action Grounding** (Translation) $\rightarrow$ **Spatiotemporal Reasoning** (Dynamics) $\rightarrow$ **Strategic Planning** (Optimization). This distinguishes our tasks from passive VQA by ensuring every question serves a functional decision-making purpose.

*   **Quality of Hard Negatives (Reviewers `Kbzn`, `MnrT`): Verified high quality.**
    We clarified our pipeline, which uses automated geometric filtering for grounding and VLM-assisted generation for semantic tasks. A rigorous manual audit of a stratified subset revealed a **96% accuracy rate** for the automated labels, ensuring that distractors are plausible but unambiguously incorrect.

We believe these additional experiments and clarifications address the core concerns regarding the validity and utility of AgentVQA. We have updated the manuscript accordingly.

Sincerely,

The Authors

---

### Meta-Review · Area_Chair_qaKT · 2026-01-07

**Summary:**

The paper introduces AgentVQA, a benchmark designed to evaluate the “agentic” capabilities of VLMs across five domains. It aggregates 14 existing datasets into a unified format, generating “hard negatives” to increase task difficulty. The authors evaluate 15 VLMs, report that top models achieve around 60% accuracy, and show that rankings on AgentVQA differ from general-purpose VQA benchmarks.

Reviewer concerns included:
(1) Lack of definition of agentic VLM / agentic capabilities.
(2) Unsubstantiated claims that prior benchmarks cannot evaluate across tasks.
(3) Lack of technical / novel contribution of methods, models, or evaluations.
(4) Unclear motivation for particular dataset selection and whether it is sufficient to capture full range of "agentic AI" capabilities.
(5) Robotic tasks focus on perception but not planning / control.
(6) Lack of verification of annotations.
(7) Oversimplification of tasks through multiple choice reformulation.
(8) Limited connection to online agent performance (e.g. latency, exploration, tool use, error recovery).

**Reviewer Concerns:**

Authors contended that the definition of agentic capabilities are broadly defined in the community with sufficient consensus, and explained their choice of datasets and motivation; they argue that technically unifying 14 fragmented datasets into a single spatial action space creates shared primitives like Grounding and Spatial Reasoning and reveals generalization gaps such as strong digital vs. weak physical grounding that are invisible to isolated benchmarks.

Authors acknowledge the lack of robotic task emphasis in planning and control (5).

Authors provide verification of a subset of annotations and provide statistics (6).

Authors describe their motivation to transform tasks to MC (7), describing this as a deliberate design to prioritize reliability, reproducibility, and scalability over the variability inherent in open-ended evaluation. The authors reasonably claim that open-ended evaluation is fundamentally incompatible with offline trajectory datasets; if a model generates a valid action different from the recorded ground truth, there is no simulation environment to generate the resulting state, rendering the prediction unverifiable. By constraining the output space, the authors ensure valid comparisons against the recorded trajectory without needing a live simulator.

Authors are currently conducting experiments to address online performance (8), which appears to be out of scope of this paper.

**Reviewer Scores:**

mHjd would likely not change their score (2); authors acknowledged and replied to all comments with explanation, but without significant changes to the manuscript in accordance to comments.
MnrT would likely not change their score (6); their concern about additional robotic tasks is acknowledged but unresolved.
KFbd would likely not change their score (4); while the MC formulation is addressed, this seemed a significant concern to this reviewer.
Kbzn would likely not change their score (4); requested experiments and analysis of online performance is left out of the scope of the paper.

---

### Decision · Program_Chairs · 2026-01-26

Reject